# Disease-relevant upregulation of P2Y$_1$ receptor in astrocytes enhances neuronal excitability via IGFBP2

Eiji Shigetomi [1,2,10] ✉, Hideaki Suzuki[1,2,10], Yukiho J. Hirayama[1], Fumikazu Sano[1,2,3], Yuki Nagai[1,2], Kohei Yoshihara[4], Keisuke Koga [4,5], Toru Tateoka[6], Hideyuki Yoshioka [6], Youichi Shinozaki [1,2], Hiroyuki Kinouchi[6], Kenji F. Tanaka [7], Haruhiko Bito [8], Makoto Tsuda [4,9] & Schuichi Koizumi [1,2] ✉

Reactive astrocytes play a pivotal role in the pathogenesis of neurological diseases; however, their functional phenotype and the downstream molecules by which they modify disease pathogenesis remain unclear. Here, we genetically increase P2Y$_1$ receptor (P2Y1R) expression, which is upregulated in reactive astrocytes in several neurological diseases, in astrocytes of male mice to explore its function and the downstream molecule. This astrocyte-specific P2Y1R overexpression causes neuronal hyperexcitability by increasing both astrocytic and neuronal Ca$^{2+}$ signals. We identify insulin-like growth factor-binding protein 2 (IGFBP2) as a downstream molecule of P2Y1R in astrocytes; IGFBP2 acts as an excitatory signal to cause neuronal excitation. In neurological disease models of epilepsy and stroke, reactive astrocytes upregulate P2Y1R and increase IGFBP2. The present findings identify a mechanism underlying astrocyte-driven neuronal hyperexcitability, which is likely to be shared by several neurological disorders, providing insights that might be relevant for intervention in diverse neurological disorders.

Astrocytes make up approximately 30% of the brain and support neuronal circuits. Emerging evidence suggests that astrocytes regulate synapses by maintaining the extracellular milieu and releasing synapse-regulating molecules[1,2], thus regulating neuronal circuits and behaviors[3–5]. Astrocytes express receptors for neurotransmitters, thereby inducing intracellular Ca$^{2+}$ signaling pathways such as those of Ca$^{2+}$ and cyclic AMP. These signaling pathways in astrocytes play a pivotal role in their interactions with neurons[6–8]. Astrocytes express several G$_q$-protein coupled receptors (G$_q$-GPCR), whose activation causes Ca$^{2+}$ elevation[9].

Astrocytes undergo massive transcriptional and morphological remodeling in response to brain injury, infection, and disease, becoming reactive astrocytes[10]. Reactive astrocytes are highly heterogeneous and contribute to the pathogenesis of neurological disorders[11–13]. Although astrocyte reactivity can be detected by several makers and the morphological appearance, functional phenotypes of

[1]Department of Neuropharmacology, Interdisciplinary Graduate School of Medicine, University of Yamanashi, Yamanashi 409-3898, Japan. [2]Yamanashi GLIA center, University of Yamanashi, Yamanashi 409-3898, Japan. [3]Department of Pediatrics, Faculty of Medicine, University of Yamanashi, Yamanashi 409-3898, Japan. [4]Department of Molecular and System Pharmacology, Graduate School of Pharmaceutical Sciences, Kyushu University, Fukuoka 812-8582, Japan. [5]Department of Neurophysiology, Hyogo College of Medicine, Hyogo 663-8501, Japan. [6]Department of Neurosurgery, Interdisciplinary Graduate School of Medicine, University of Yamanashi, Yamanashi 409-3898, Japan. [7]Division of Brain Sciences, Institute for Advanced Medical Research, Keio University School of Medicine, Tokyo 160-8582, Japan. [8]Department of Neurochemistry, Graduate School of Medicine, The University of Tokyo, Tokyo 113-0033, Japan. [9]Department of Life Innovation, Graduate School of Pharmaceutical Sciences, Kyushu University, Fukuoka 812-8582, Japan. [10]These authors contributed equally: Eiji Shigetomi, Hideaki Suzuki. ✉e-mail: eshigetomi@yamanashi.ac.jp; skoizumi@yamanashi.ac.jp

the reactive astrocytes are still unclear[14,15]. Reactive astrocytes generally display aberrant $Ca^{2+}$ signals, a proxy of astrocyte function, which contributes to disease pathogenesis and could be therapeutic targets[16–21]. However, aberrant $Ca^{2+}$ signals do not always reflect the function of reactive astrocytes. $Ca^{2+}$ signals in reactive astrocytes either increase or decrease and the neuronal impacts vary depending on the disease type, stage, age, and other variables[20]. In addition, $Ca^{2+}$ signals in astrocytes are not binary signals for neuronal impacts[22]. Therefore, to elucidate the molecular mechanism underlying aberrant $Ca^{2+}$ signals and the link between aberrant $Ca^{2+}$ signals and the downstream molecules is crucial for a better understanding of functional phenotypes of reactive astrocytes.

Of the molecules that contribute to aberrant $Ca^{2+}$ signals in astrocytes, some $G_q$-GPCRs, such as the $P2Y_1$ receptor (P2Y1R) and metabotropic glutamate receptor 5 (mGluR5), are upregulated in multiple disease models; their upregulation may impact disease pathogenesis[20,23,24]. Among these $G_q$-GPCRs, P2Y1R is upregulated and/or activated in astrocytes in several neurological disease models, such as Alzheimer's disease (AD)[25,26], epilepsy[27–30], stroke[31,32], and traumatic brain injury[33]. It has also been suggested that P2Y1R upregulation contributes to disease pathogenesis[34]. However, it remains unknown whether reactive astrocytes with upregulated P2Y1R share a common functional phenotype in neurological diseases[14]. In addition, although the pharmacological blockade of P2Y1R ameliorates several aspects of disease phenotypes[26,31–33], pharmacological methods do not specifically target astrocytes, which may cause side effects when treating disease pathogenesis[17]. Elucidating the precise downstream pathway of astrocytes for their interaction with neurons is therefore crucial for understanding the role of P2Y1R-expressing reactive astrocytes.

In the present study, we utilized a method for the astrocyte-specific overexpression of P2Y1R, without affecting injuries or infections, using a Tet-Off conditional transgenic mice line (astrocyte-specific P2Y1R overexpression [AstroP2Y1OE] mice). We then investigated the effects of this overexpression on neurons, circuits, and behavior. Using adult male mice, we performed brain slice imaging, electrophysiology, RNA sequencing (RNA-seq) analysis, histology, electroencephalogram (EEG) recordings, and behavioral analysis. P2Y1R overexpression enhanced neuron–astrocyte interaction and upregulated insulin-like growth factor-binding protein 2 (IGFBP2), an "excitatory signal" that enhances neuronal excitation through IGF-1R, leading to neuronal hyperexcitability and enhanced seizure susceptibility. Our data indicate that the functional phenotype of reactive astrocytes with upregulated P2Y1R-IGFBP2 signaling is shared in several neurological diseases, and may contribute to disease pathogenesis.

## Results

### Neuronal hyperexcitability is induced by astrocytic P2Y1R overexpression

To achieve astrocytic P2Y1R overexpression, we used a Tet-OFF system. We crossed an Mlc1-tTA line (an astrocytic tTA line[35,36]) with a P2Y1tetO line. We have previously reported that *P2ry1* transcripts are highly upregulated in GFAP-positive astrocytes of AstroP2Y1OE by in situ hybridization[37]. To confirm that the protein levels are also upregulated, we performed immunohistochemistry. Indeed, P2Y1R immunofluorescence was significantly higher in GFAP-positive astrocytes in the hippocampal CA1 region of AstroP2Y1OE mice (Supplementary Fig. 1A). We started by investigating whether P2Y1R overexpression in astrocytes affects neuronal excitability. We first recorded spontaneous neuronal activities by placing a bipolar electrode into the hippocampus of freely behaving AstroP2Y1OE and control mice. We focused on the spikes that resembled those observed in epilepsy model mice[38]. The frequency of spikes was significantly higher in AstroP2Y1OE mice than in control mice (Fig. 1A). At the end of the experiments, we administered pilocarpine to induce epileptic seizures and recorded abundant neuronal spikes (Supplementary Fig. 1B). These findings

suggest that astrocytic P2Y1R overexpression leads to neuronal hyperexcitability. Next, we tested whether AstroP2Y1OE mice had increased seizure susceptibility. When we administered pilocarpine to induce drug-induced status epilepticus (Racine scale 5), the cumulative dose of pilocarpine was lower in AstroP2Y1OE mice than in control mice (Fig. 1B), indicating that AstroP2Y1OE mice increased seizure susceptibility. For insights into neuronal hyperexcitability at a cellular level, we performed patch-clamp recordings from granule neurons in the dentate gyrus using acute brain slices. Neurons from AstroP2Y1OE mice had higher firing frequencies than those from control mice in response to positive current injections (Supplementary Fig. 2A–E), but there were no differences in resting membrane potential, input resistance, or membrane capacitance. Together, these findings suggest that astrocytic P2Y1R overexpression leads to neuronal hyperexcitability.

### Dual-color $Ca^{2+}$ imaging of neurons and astrocytes

Previously, we found that astrocyte-specific P2Y1R overexpression increased $Ca^{2+}$ signals in astrocytes (i.e., there were larger responses to electrical field stimulation [EFS] and spontaneous $Ca^{2+}$ waves)[37]. Because $Ca^{2+}$ signals in astrocytes can affect neuronal excitability by triggering the release of neuroactive substances (i.e., gliotransmitters)[2], we hypothesized that increased bidirectional interactions between neurons and astrocytes might occur, thus causing neuronal hyperexcitability. To analyze bidirectional communication between astrocytes and neurons, we injected a mixture of adeno-associated viruses (AAVs) into the dorsal hippocampus of either AstroP2Y1OE or control mice to express GCaMP6f[39] (a green fluorescent genetically encoded $Ca^{2+}$ indicator) and jRGECO1a[40] (a red fluorescent genetically encoded $Ca^{2+}$ indicator) into astrocytes and neurons, respectively, to visualize $Ca^{2+}$ signals in both cell types simultaneously. Three to four weeks after injection, we cut acute brain slices and measured $Ca^{2+}$ signals in both neurons and astrocytes in the CA1 region of the hippocampus under a two-photon microscope. The Schaffer collaterals were stimulated at 40 Hz to observe evoked $Ca^{2+}$ responses. We set the region of interest (ROI) as the stratum radiatum. Everywhere in the field of view, neuronal $Ca^{2+}$ increased immediately after the stimuli began. Astrocytic $Ca^{2+}$ responses in AstroP2Y1OE mice were much larger than those of control mice and covered entire astrocytic territories (Fig. 1C, Supplementary Videos 1–3). In control mice, astrocytic $Ca^{2+}$ responses were highly localized and did not propagate to other regions of astrocytes. In the field of view (approximately 300 μm away from the stimulation electrode), the evoked $Ca^{2+}$ signals in control mice were very small, which is consistent with our previous observation[37]. The lack of clear astrocytic $Ca^{2+}$ signals in the control mice is somewhat inconsistent with previous reports of astrocytic $Ca^{2+}$ signals in the same region of the hippocampus[41,42]. This discrepancy is likely caused by differences in imaging/stimulus settings; presumably, we used a weaker stimulus intensity than past reports, meaning that astrocytic $Ca^{2+}$ signals in the control mice were very small. Astrocytic $Ca^{2+}$ responses were a few seconds delayed in AstroP2Y1OE mice compared with controls. Astrocytic $Ca^{2+}$ responses in AstroP2Y1OE mice generally started at the processes, where P2Y1Rs are presumably abundant[37]. Neuronal $Ca^{2+}$ responses increased linearly as the stimulus number increased, whereas astrocytic $Ca^{2+}$ responses reached a peak with a stimulus number of around 100 (Fig. 1D–F). Both neuronal and astrocytic $Ca^{2+}$ signals were larger in AstroP2Y1OE mice, suggesting an increase in bidirectional communication between astrocytes and neurons in AstroP2Y1OE mice. Basal excitatory synaptic transmission was not altered when we measured fEPSP slopes (Supplementary Fig. 2F–H) and evoked EPSC amplitude (Supplementary Fig. 2J). The paired-pulse ratio was lower in AstroP2Y1OE (Supplementary Fig. 2K), suggesting the increase in glutamate release probability by AstroP2Y1OE. Overall, neuronal $Ca^{2+}$ signals in AstroP2Y1OE were larger in response to repetitive inputs of excitatory synapses (Fig. 1D–F).

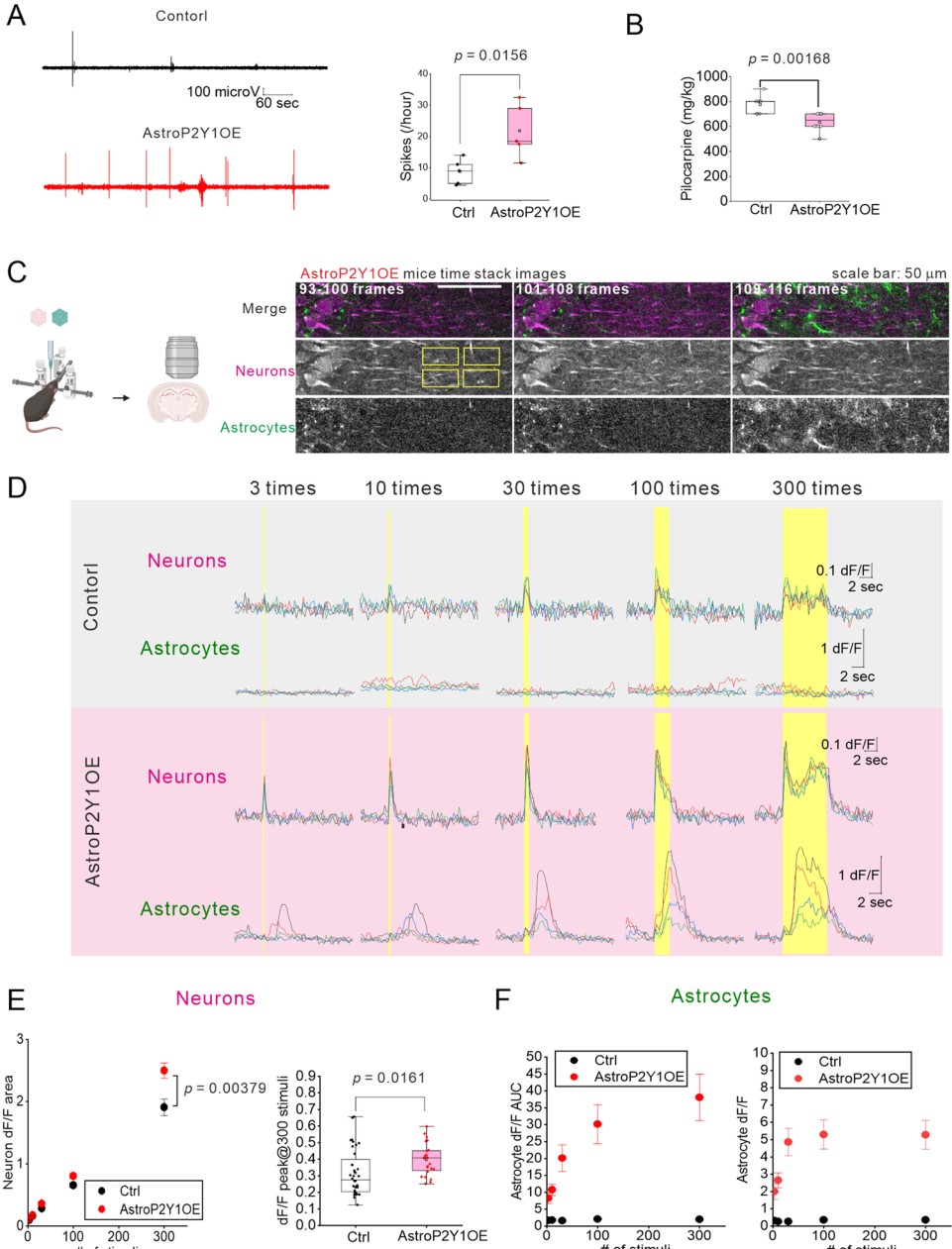

**Fig. 1 | Astrocytic P2Y1R overexpression leads to decreased susceptibility to drug-induced seizures, increased abnormal spikes, and increased bidirectional communication between neurons and astrocytes. A** Representative EEG traces. A bipolar electrode was placed in the left hippocampus. The AstroP2Y1OE mice showed more spikes. Control, N = 5 mice; AstroP2Y1OE, N = 5 mice (two-sided two-sample t-test). Box-plot elements are defined in the following way: center line, median; box limits, upper and lower quartiles; whiskers, 1.5× interquartile range; square, mean. **B** Mice were administrated pilocarpine (100 mg/kg) every 20 min to reach severe seizure (Racine stage 5). The bar graph indicates the cumulative dose of pilocarpine needed to induce seizure at Racine stage 5. Control, N = 8 mice; AstroP2Y1OE, N = 12 mice (two-sided Mann–Whitney U test). Box-plot elements are defined in the following way: center line, median; box limits, upper and lower quartiles; whiskers, 1.5× interquartile range; square, mean. **C** Left, Schematic diagram for the AAV-mediated expression of GCaMP6f in astrocytes and jRGECO1a in neurons. The schematic diagram in (**C**) was created with BioRender.com released under a Creative Commons Attribution-NonCommercial-NoDerivs 4.0 International license. Right, Representative images of dual-color Ca²⁺ imaging of astrocytes (GcaMP6f) and neurons (jRGECO1a) in the stratum radiatum of the hippocampal CA1 region in AstroP2Y1OE mice. Time-stacked images were obtained from eight frames, as indicated at the top of the images (AstroP2Y1OE). Images were taken every 300 ms. The Schaffer collaterals were stimulated by an electrode (0.06 mA,

40 Hz, 2.5 s) between 101 and 108 frames. Yellow rectangles in the left panel indicate the ROIs. The same experiments were conducted using 9 slices, 4 mice for control and 6 slices, 4 mice for AstroP2Y1OE. **D** Sample traces of EFS-evoked Ca²⁺ signals. Both neuronal and astrocytic Ca²⁺ signals were enhanced in response to the EFS in AstroP2Y1OE mice. EFS was delivered at the duration indicated by yellow boxes. **E** Summary of EFS-evoked Ca²⁺ signals for neurons. Neuronal Ca²⁺ responses were enhanced in AstroP2Y1OE mice (red circles) compared with the control (black circles). The left panel shows the data for Ca²⁺ responses as the area under the curve. The right panel shows the data for peak Ca²⁺ responses. Control, n = 36 ROIs from 9 slices, 4 mice; AstroP2Y1OE, n = 24 ROIs from 6 slices, 4 mice (two-way repeated measures ANOVA and Tukey's test for the left panel; two-sided two-sample t-test for the right panel). For the left panel, data are presented as mean ± s.e.m. For the right panel, box-plot elements are defined in the following way: center line, median; box limits, upper and lower quartiles; whiskers, 1.5× interquartile range; square, mean. **F** Summary of EFS-evoked Ca²⁺ signals for astrocytes. ROIs were set at the same regions of FOVs of GCaMP6f images so that n numbers are the same as those in (**E**). The left panel shows the data for Ca²⁺ signals as the area under the curve. The right panel shows the data for peak Ca²⁺ signals. Data are presented as mean ± s.e.m. Black circles, control; red circles, AstroP2Y1OE. Source data are provided as a Source Data file.

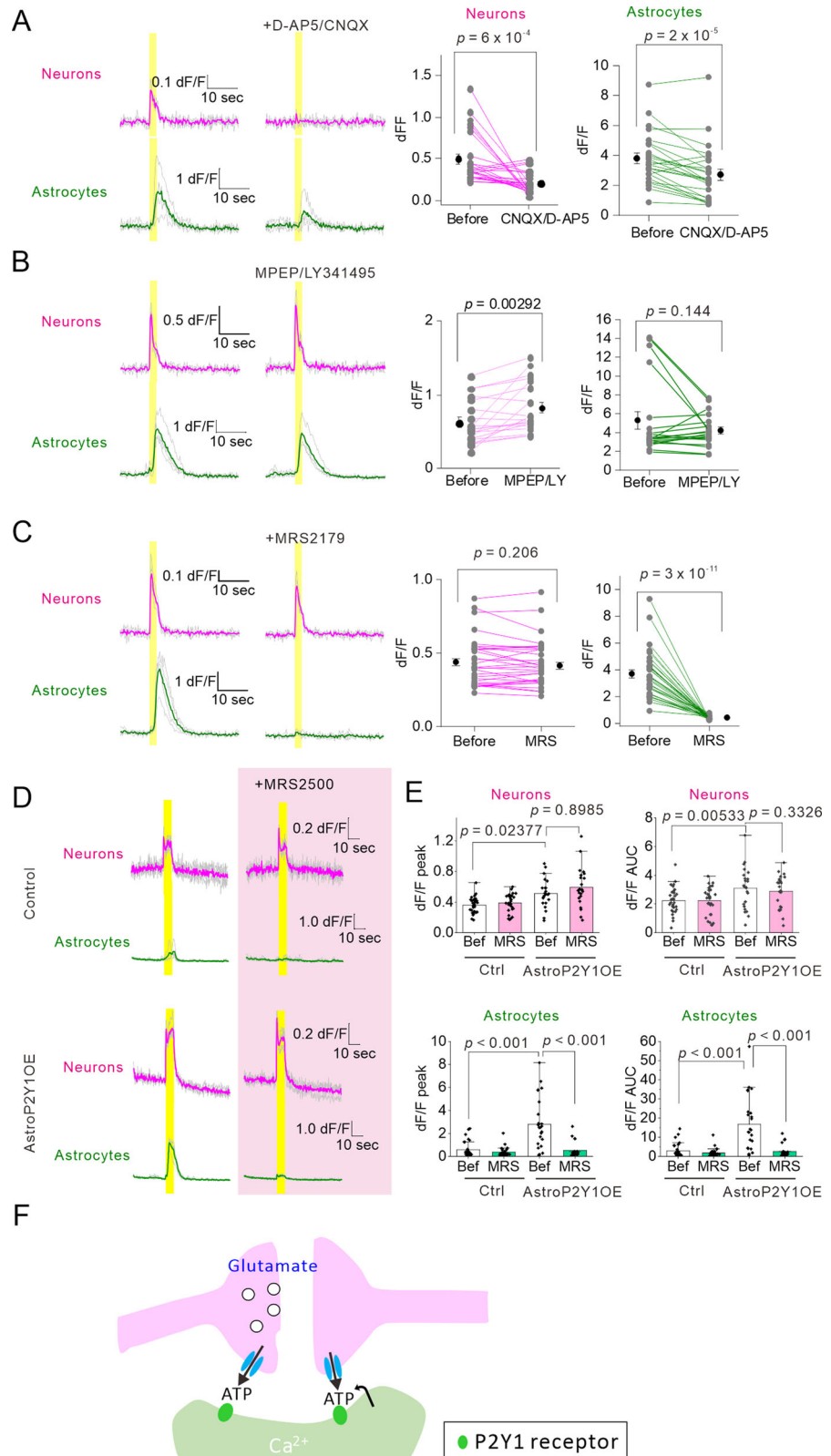

To determine the neurotransmitters and neuromodulators that induce Ca²⁺ signals in each of the two cell types, we performed pharmacological experiments in the presence of picrotoxin to block inhibitory synaptic transmission. Both neuronal and astrocytic Ca²⁺ responses were blocked by tetrodotoxin (TTX) (Supplementary Fig. 3A, B), suggesting that they occur via action-potential-dependent neurotransmitter release. Furthermore, neuronal Ca²⁺ signals were blocked by cocktails of the ionotropic glutamate receptor antagonists cyanquixaline (CNQX) and D-2-amino-5-phosphonopentanoate (D-AP5) (Fig. 2A), indicating that glutamate mediates neuronal Ca²⁺ signals. Unlike neuronal Ca²⁺ signals, astrocytic Ca²⁺ signals remained but were significantly reduced in the presence of CNQX/D-AP5, suggesting that both presynaptic and postsynaptic activities are required for astrocytic Ca²⁺ signals (Fig. 2A). The blockade of NMDA receptors with

**Fig. 2 | Pharmacological analysis of EFS-evoked Ca²⁺ signals in astrocytes in AstroP2Y1OE mice. A** Sample traces of Ca²⁺ signals. CNQX (10 μM)/D-AP5 (50 μM) abolished neuronal Ca²⁺ signals evoked by EFS (40 Hz, 100 times, 0.06 mA) in AstroP2Y1OE mice. Astrocytic Ca²⁺ signals were reduced in the presence of CNQX/ D-AP5 but remained present. n = 28 ROIs from 8 slices, 6 mice (two-sided paired t-test). The experiments were performed in the presence of picrotoxin (100 μM). **B** Sample traces of Ca²⁺ signals. MPEP (50 μM)/LY341495 (10 μM) did not reduce neuronal or astrocytic Ca²⁺ signals evoked by EFS (40 Hz, 100 times, 0.06 mA) in AstroP2Y1OE mice. n = 28 ROIs from 8 slices, 3 mice (two-sided paired t-test). The experiments were performed in the presence of picrotoxin (100 μM). **C** Summary of EFS (40 Hz, 100 times, 0.06 mA)-evoked Ca²⁺ signals in control mice before and during the MRS2179 application. MRS2179 abolished astrocytic Ca²⁺ signals but did not affect peak Ca²⁺ signals in AstroP2Y1OE mouse neurons. n = 32 ROIs from

9 slices, 6 mice (two-sided paired t-test). The experiments were performed in the presence of picrotoxin (100 μM). **D** The effects of MRS2500 (1 μM) on EFS (40 Hz, 300 times, 0.06 mA)-evoked Ca²⁺ signals in neurons and astrocytes. **E** Summary of the effects of MRS2500 on neuronal and astrocytic Ca²⁺ signals in control mice. Control, n = 32 ROIs, 8 slices, 3 mice; AstroP2Y1OE, n = 24 ROIs, 6 slices, 3 mice (one-way ANOVA and Tukey's test). MRS2500 inhibited astrocytic Ca²⁺ signals; however, it did not affect neuronal Ca²⁺ signals in AstroP2Y1OE mice. Note that neuronal Ca²⁺ signals were larger in AstroP2Y1OE mice than in control mice. The experiments were performed in the presence of picrotoxin (100 μM). Data are presented as mean ± s.d. **F** The schematic shows neuromodulators and their sources that cause Ca²⁺ signals in neurons and astrocytes. Source data are provided as a Source Data file.

D-AP5 reduced neuronal but not astrocytic Ca²⁺ signals (Supplementary Fig. 3C), suggesting that NMDA-receptor activation may not play a major role in astrocytic Ca²⁺ signals in our experimental conditions. The presence of NMDA-receptor-independent component of neuronal Ca²⁺ signals indicates the contribution of voltage-gated Ca²⁺ channels in the neuronal Ca²⁺ signals. On the other hand, cocktails of 2-methyl-6-(phenylethynyl)pyridine (MPEP), a mGluR5 receptor antagonist, and LY341495, an mGluR2/3 receptor antagonist, did not affect astrocytic Ca²⁺ signals but increase neuronal Ca²⁺ signals (Fig. 2B). The P2Y1R antagonists MRS2179 and MRS2500 almost abolished Ca²⁺ signals in astrocytes, suggesting that P2Y1R mediates Ca²⁺ signals in astrocytes in AstroP2Y1OE mice. Because P2Y1R mainly mediates astrocytic Ca²⁺ signals in AstroP2Y1OE, we wondered whether ATP release in response to neuronal activities might be increased in AstroP2Y1OE. Then, we performed extracellular ATP imaging using GRAB$_{ATP1.0}$, a genetically encoded ATP sensor[43], into astrocytes by AAV. Although the area under the curve of the responses was slightly increased in Astro-P2Y1OE, the peak responses were equivalent (Supplementary Fig. 4), suggesting that larger Ca²⁺ signals in astrocytes of AstroP2Y1OE are mainly due to the upregulation of P2Y1R molecules, which enable to respond to the ATP release more effectively.

Contrary to the effect on astrocytic Ca²⁺ signals, P2Y1R antagonists did not affect the peak of neuronal Ca²⁺ signals (Fig. 2C–E). This result indicates that enhanced neuronal Ca²⁺ signals in AsroP2Y1OE are not induced by the instantaneous release of gliotransmitters upon P2Y1R activation and the subsequent increased Ca²⁺ signals, but rather by a relatively slow mechanism. Together, these findings suggest that astrocytic Ca²⁺ signals in AstroP2Y1OE mice are mainly mediated by ATP (or ADP) in response to neuronal activity, whereas enhanced neuronal excitation in AstroP2Y1OE mice is mediated by the activity-dependent release of glutamate which is presumably derived from presynaptic terminals (Fig. 2F).

Several pathways for ATP release have been identified in neurons and astrocytes, including channels and exocytotic pathways[44]. To elucidate the ATP/ADP release pathway contributing to astrocytic Ca²⁺ signals, we aimed to pharmacologically block some of the ATP release pathways. Carbenoxolone (CBX) and probenecid, which block connexins and pannexins, significantly reduced astrocytic Ca²⁺ signals. CBX did not affect neuronal Ca²⁺ signals. However, probenecid reduced neuronal Ca²⁺ signals. The anion channel blocker 5-nitro −2-(3-phenylpropylamino) benzoic acid (NPPB) also inhibits pannexins and this led to reduced astrocytic Ca²⁺ signals and partially reduced neuronal Ca²⁺ signals. Neither 4-(2-butyl-6,7-dichloro-2-cyclopenty-lindan-1-on-5-yl) oxobutyric acid (DCPIB), a volume-regulated anion channel blocker, nor chlodronate, a vesicular nucleotide transporter inhibitor, affected astrocytic Ca²⁺ signals. However, DCPIB increased neuronal Ca²⁺ signals. Although the pharmacological reagents that were used are not entirely specific, the overall pharmacological data suggest that some ATP may be released through pannexin/connexin hemichannels expressed in neurons and/or astrocytes (Supplementary Fig. 3D, E)[45–47].

## Glutamatergic transmission is enhanced in astrocytes overexpressing P2Y1Rs

Enhanced neuronal Ca²⁺ responses were mediated by ionotropic glutamatergic receptors, as shown in Fig. 2. We next investigated whether P2Y1R upregulation enhances glutamate release in AstroP2Y1OE mice[27,48–50]. We imaged extracellular glutamate levels using the glutamate sensor iGluSnFR[51,52] in astrocytes. Stimuli-evoked iGluSnFR signals were significantly enhanced (Fig. 3A–D), suggesting that astrocytic P2Y1R overexpression increases glutamate availability. Moreover, iGluSnFR signals returned to baseline after stimulation, whereas astrocytic Ca²⁺ signals slowly increased and peaked after stimulation. These findings suggest that elevated glutamate is likely derived from neurons rather than astrocytes. Because basal excitatory synaptic transmission was not altered but the glutamate release probability was enhanced in AstroP2Y1OE (Supplementary Fig. 2), the enhanced glutamate iGluSnFR signal is probably due to an increase in glutamate release from presynaptic terminals. To test whether astrocytes directly contributed to glutamate elevation in our experiments, we performed a puff application of MRS2365. This treatment increased Ca²⁺ but did not elevate glutamate near astrocytes (Fig. 3E), indicating that the instantaneous activation of P2Y1Rs in astrocytes does not elevate glutamate levels under our experimental conditions. However, we cannot rule out the possibility of Ca²⁺-dependent glutamate release from astrocytes[53]. Overall, these findings suggest that P2Y1R-overexpressing astrocytes regulate glutamatergic signals partly by facilitating glutamate release from presynaptic terminals, although other mechanisms to enhance glutamatergic transmission may also play a role (see "Discussion" section).

## Gene expression profile of P2Y1R-overexpressing astrocytes

We hypothesized that P2Y1R-overexpressing astrocytes might upregulate an unknown excitatory signal to enhance glutamatergic signals in neurons. To elucidate the excitatory signal derived from astrocytes, we performed RNA-seq on hippocampal astrocytes that were isolated by magnetic-activated cell sorting (Fig. 4A). All the gene expression data are in Supplementary Data 1. Cell-type-specific genes confirmed that the extracted RNA was mainly derived from astrocytes and not neurons, microglia, or oligodendrocytes, although there was a small amount of contamination of endothelial genes (Fig. 4B). Interestingly, reactive astrocyte markers including glial fibrillary acidic protein (*Gfap*) were not upregulated (Fig. 4C), suggesting that the cell dissociation procedure causes minimal damage to astrocytes and that P2Y1R overexpression does not alter astrocytic reactivity; this is consistent with the results of our previous immunohistochemical analysis of GFAP[37]. We set differentially expressed genes (DEGs) as those with false discovery rate <0.05 and fragments per kilobase of transcript per million mapped reads (FPKM) >1 and revealed that 17 DEGs were upregulated and 27 DEGs were downregulated (Fig. 4D–F). As expected, *P2ry1* was the most upregulated gene (Fig. 4D). Other genes related to purinergic signaling were not altered (Fig. 4G). Furthermore, the expression of genes related to glutamate transport and glutamate

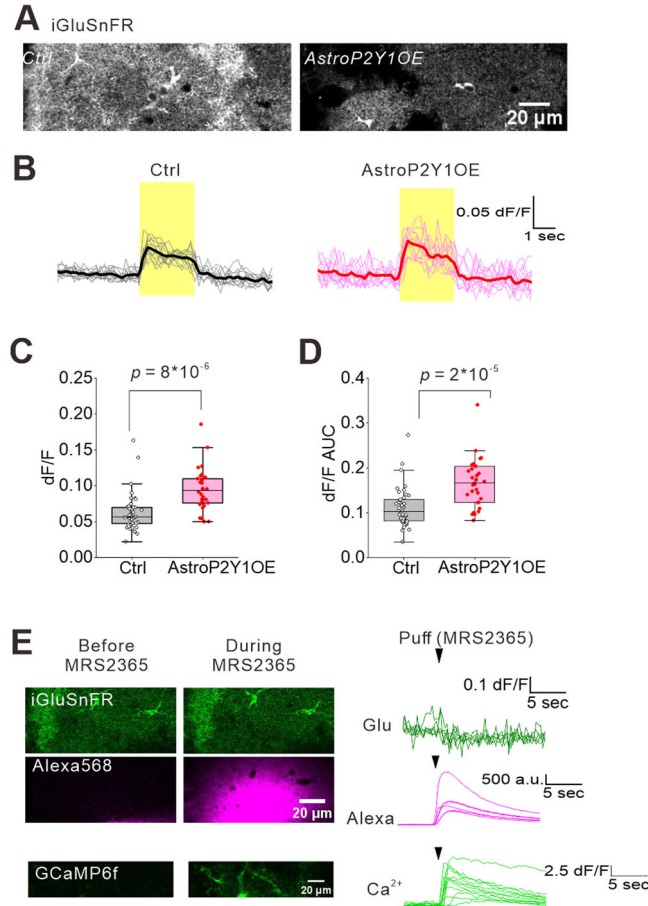

**Fig. 3 | Glutamate release is enhanced by astrocyte P2Y1R overexpression.**
**A** Sample images of iGluSnFR-expressing astrocytes. The same experiments were conducted using 11 slices, 4 mice for control and 8 slices, 3 mice for AstroP2Y1OE.
**B** Sample traces of glutamate responses evoked by EFS (40 Hz, 100 times, 0.06 mA) of the Schaffer collaterals. The experiments were performed in the presence of picrotoxin (100 $\mu$M). **C, D** Summary of EFS-evoked glutamate responses at the peak (**C**) and area under the curve (**D**). Control, n = 41 ROIs from 11 slices, 4 mice; AstroP2Y1OE, n = 30 ROIs from 8 slices, 3 mice (two-sided two-sample t-test). EFS-evoked glutamate levels were higher in AstroP2Y1OE mice. Box-plot elements are defined in the following way: center line, median; box limits, upper and lower quartiles; whiskers, 1.5×× interquartile range; square, mean. **E** A puff application of MRS2365 (10 $\mu$M, 5 psi, 500 ms) did not alter iGluSnFR signals but induced very large Ca$^{2+}$ signals in astrocytes. n = 4 slices, 2 mice for each imaging experiment. Source data are provided as a Source Data file.

receptors in astrocytes was not altered in AstroP2Y1OE mice (Fig. 4H). Immunohistochemical analysis of GLT-1 and GLAST revealed similar expression levels in AstroP2Y1OE mice compared to control mice (Supplementary Fig. 5A, B). These data suggest that P2Y1R overexpression may increase neuronal excitation (Fig. 1) and glutamate level (Fig. 3) without affecting glutamate signaling, uptake, and/or metabolism.

### Astrocyte-derived IGFBP2 enhances neuronal excitation
Of the 17 upregulated genes (Fig. 4F), we focused on *Igfbp2* because it was the only gene that fulfilled the following criteria. First, *Igfbp2* is enriched in astrocytes[12,54,55]. Second, *Igfbp2* encodes a secretory protein; in a proteomic study, IGFBP2 was found in astrocyte-conditioned medium from hippocampal astrocyte cultures[56]. Third, IGFBP2 alters neuronal excitability; past studies have reported that IGFBP2 can enhance hippocampal glutamatergic synaptic transmission and firing in development[57] and is an aberrant astrocytic protein in developmental disorders[58].

Because IGFBP2 is highly expressed in development and its expression declines in adults, we first confirmed that IGFBP2 protein levels were higher in astrocytes in AstroP2Y1OE mice compared with controls. We performed immunohistochemistry against IGFBP2 and GFAP. IGFBP2 was colocalized with GFAP in the stratum radiatum of the hippocampal CA1 region and the molecular layer of the dentate gyrus. The IGFBP2 signals that colocalized with GFAP were stronger in AstroP2Y1OE mice than in control mice (Fig. 5A, B and Supplementary Fig. 5C–E). GFAP signals were equivalent between control and AstroP2Y1OE mice, which was consistent with the RNA-seq data. These results indicate that astrocytes have increased IGFBP2 levels in AstroP2Y1OE mice.

Next, we investigated whether IGFBP2 acts as an excitatory signal. To inhibit the action of IGFBP2, we treated acute slices with IGFBP2-neutralizing antibodies for 2 h. This treatment reduced neuronal Ca$^{2+}$ signals in AstroP2Y1OE mice compared with IgG control treatment but did not affect astrocytic Ca$^{2+}$ signals in AstroP2Y1OE mice (Fig. 5C). To determine whether IGFBP2 itself increases neuronal excitation, we next treated slices with recombinant IGFBP2 (10 ng/mL) for 1 h. This treatment enhanced Ca$^{2+}$ signals in the soma (Fig. 5D). The effect of the recombinant IGFBP2 was weaker at dendritic regions of neurons than soma but significantly larger at the first phase of evoked Ca$^{2+}$ signals than that of control (Supplementary Fig. 6). To determine the effects of astrocyte-derived IGFBP2 on neuronal Ca$^{2+}$ signals, we performed gene knockdown using the AAV-mediated clustered regularly interspaced palindromic repeats (CRISPR)/Cas9 system[59]. We expressed *Staphylococcus aureus* Cas9 (SaCas9) under the GfaABC$_1$D promoter and also expressed single-guide RNA targeting exon 2 of *Igfbp2* (Fig. 5E). Approximately 63% of SOX9-positive astrocytes expressed SaCas9 and mCherry reporter for the single-guide RNA (Supplementary Fig. 7A). We confirmed a decrease (approximately 30%) in IGFBP2 expression in SOX9-positive astrocytes using immunohistochemistry compared with controls (Supplementary Fig. 7B). To investigate the effects of genome editing in astrocytes on neuronal Ca$^{2+}$ signals, we used an AAV vector to express jGCaMP8s[60] under the synapsin promoter. Knockdown of *Igfbp2* reduced Schaffer-collateral-stimulation-induced Ca$^{2+}$ signals in the soma of neurons in AstroP2Y1OE mice (Fig. 5F). Furthermore, the knockdown decreased the first phase of Ca$^{2+}$ signals at dendrites (Supplementary Fig. 7C, D). Together, these results indicate that the enhancement of neuronal Ca$^{2+}$ signals in AstroP2Y1OE mice is mediated by astrocyte-derived IGFBP2. Several targets for IGFBP2 have been proposed[57,61]. We, therefore, conducted a pharmacological analysis to explore the mechanisms of the IGFBP2-induced enhancement of neuronal excitation that was observed in AstroP2Y1OE mice. Pharmacological blockade of the IGF-1R reduced both the peak and the area under the curve of neuronal Ca$^{2+}$ signals in AstroP2Y1OE mice but not in control mice (Fig. 6A, B). Neither IGF-2R nor IGF-1 blockade affected neuronal Ca$^{2+}$ signals in AstroP2Y1OE mice (Fig. 6C, D). These findings suggest that the IGF-1R contributes to the IGFBP2-mediated enhancement of neuronal Ca$^{2+}$ signals in AstroP2Y1OE mice.

### In neurological disease models, IGFBP2 is co-upregulated with P2Y1R in astrocytes
Astrocytes upregulate P2Y1R in several neurological diseases such as epilepsy, stroke, and AD[25,27,31]. We, therefore, evaluated whether IGFBP2 is upregulated in astrocytes with P2Y1R upregulation by creating a kainic-acid-induced seizure model. Four days after the unilateral injection of kainic acid (20 mM), approximately 74% of astrocytes were P2Y1R positive in the contralateral CA1 region; this was much higher than in the saline control (Fig. 7A). IGFBP2 was markedly upregulated in GFAP-positive astrocytes in the contralateral CA1 region; approximately 70% of astrocytes were IGFBP2 positive. Both P2Y1R- and IGFBP2-positive astrocytes were similarly distributed in the CA1 region, suggesting that most astrocytes co-express both P2Y1R

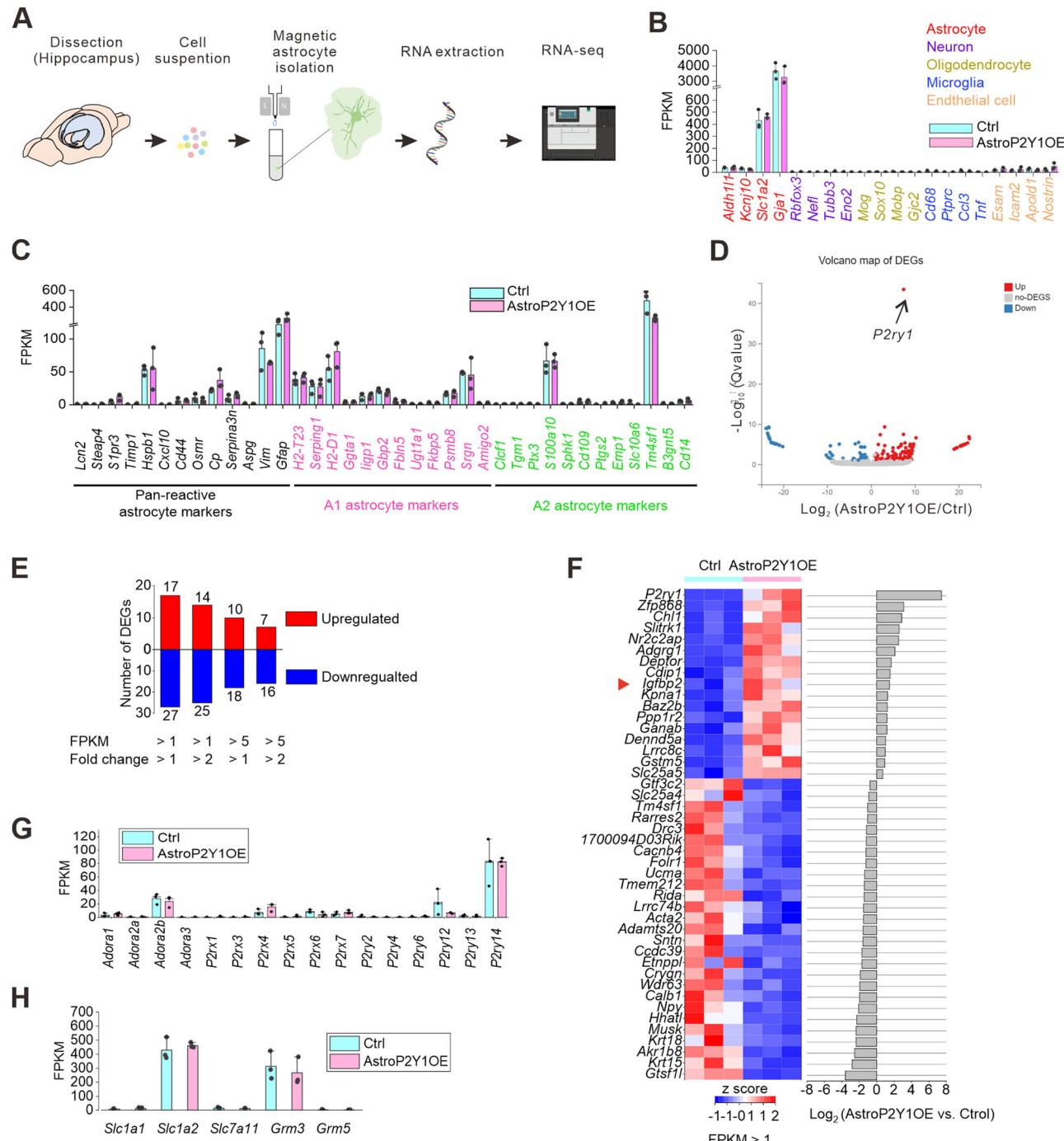

**Fig. 4 | RNA-seq analysis of AstroP2Y1OE mice. A** Schematic diagram of RNA isolation from the hippocampus. **B** Cell-type-specific RNA expression (N = 3 samples per group). Data are presented as mean ± s.d. Light blue column, control; magenta column, AstroP2Y1OE. **C** FPKM values for pan-reactive, A1, and A2 astrocyte markers (N = 3 samples per group). Data are presented as mean ± s.d. **D** Volcano plot for the RNA-seq analysis. Red circles, upregulated DEGs; gray circles, No-DEGs; blue circles, downregulated DEGs. **E** Number of DEGs in AstroP2Y1OE mice compared with control mice. Red column, upregulated DEGs; blue column, downregulated DEGs. **F** Heat map of DEGs. **G** FPKM values for genes related to purinergic signaling (N = 3 samples per group). Data are presented as mean ± s.d. Light blue column, control; magenta column, AstroP2Y1OE. **H** FPKM values for genes related to glutamate homeostasis (N = 3 samples per group). Data are presented as mean ± s.d. Light blue column, control; magenta column, AstroP2Y1OE. Source data are provided as a Source Data file.

and IGFBP2. Given that P2Y1R[31] and IGFBP2[62,63] are reportedly upregulated by ischemic insults, we wondered if there might be a correlation between these proteins in astrocytes in a stroke model. We, therefore, performed similar experiments using a stroke model caused by 15 min of middle cerebral artery occlusion (MCAO). There was P2Y1R and IGFBP2 upregulation in the striatum of the ischemic (ipsilateral) but not intact (contralateral) side 3 days after MCAO (Fig. 7B).

Approximately 47% of the GFAP-labeled astrocytes were P2Y1R positive, and approximately 24% of the astrocytes were IGFBP2 positive. Upregulation of P2Y1R and IGFBP2 is also observed in the cortex and the hippocampus (Supplementary Fig. 8). These data suggest that astrocytes expressing high levels of P2Y1R upregulate IGFBP2 in neurological disease models. We analyzed publicly available datasets whether *P2ry1/P2RY1* and *Igfbp2/IGFBP2* transcripts are upregulated in

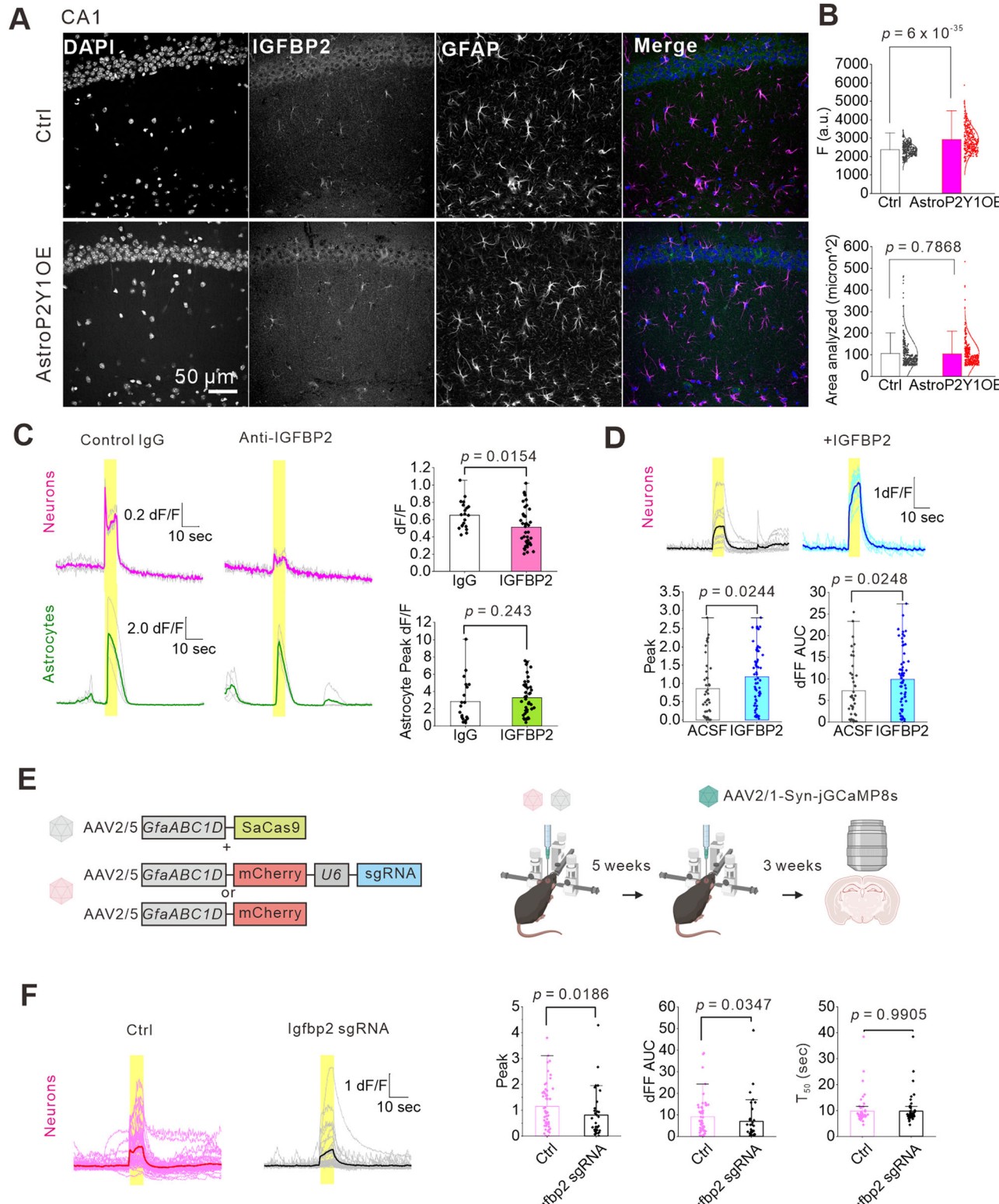

**Fig. 5 | IGFBP2 in astrocytes leads to enhanced neuronal activities in AstroP2Y1OE mice. A**, **B** Immunohistochemical analysis of IGFBP2 expression in the hippocampus. GFAP was used as an astrocytic marker. IGFBP2 signals that were colocalized with GFAP were quantified. Control, n = 293 ROIs from 15FOVs, 3 mice; AstroP2Y1OE, n = 331 ROIs from 12 FOVs, 3 mice (two-sided two-sample t-test). Data are presented as mean ± s.d. **C** Representative traces of EFS-evoked neuronal Ca²⁺ signals (magenta) and astrocytic Ca²⁺ signals (green). Compared with control IgG (n = 20 ROIs from 3 slices, 3 mice), IGFBP2 antibodies reduced neuronal Ca²⁺ signals without affecting astrocytic Ca²⁺ signals in AstroP2Y1OE mice (n = 40 ROIs from 6 slices, 3 mice) (two-sided two-sample t-test). Data are presented as mean ± s.d. **D** IGFBP2

at 10 ng/mL (1 h at room temperature) enhanced EFS-evoked neuronal Ca²⁺ signals in the soma. ACSF (control), n = 42 cells from 5 slices, 3 mice; IGFBP2, n = 61 cells from 6 slices, 3 mice (two-sided two-sample t-test). Data are presented as mean ± s.d. **E** Schematic diagram for the AAV-mediated knockdown of *Igfbp2* in astrocytes. The schematic diagram in Fig. 6E was created with BioRender.com released under a Creative Commons Attribution-NonCommercial-NoDerivs 4.0 International license. **F** Traces of neuronal Ca²⁺ signals in the soma of AstroP2Y1OE mice with or without *Igfbp2* knockdown in astrocytes. Control, n = 55 cells from 9 slices, 4 mice; *Igfbp2* gRNA, n = 33 cells from 5 slices, 3 mice (two-sided Mann–Whitney U test). Data are presented as mean ± s.d. Source data are provided as a Source Data file.

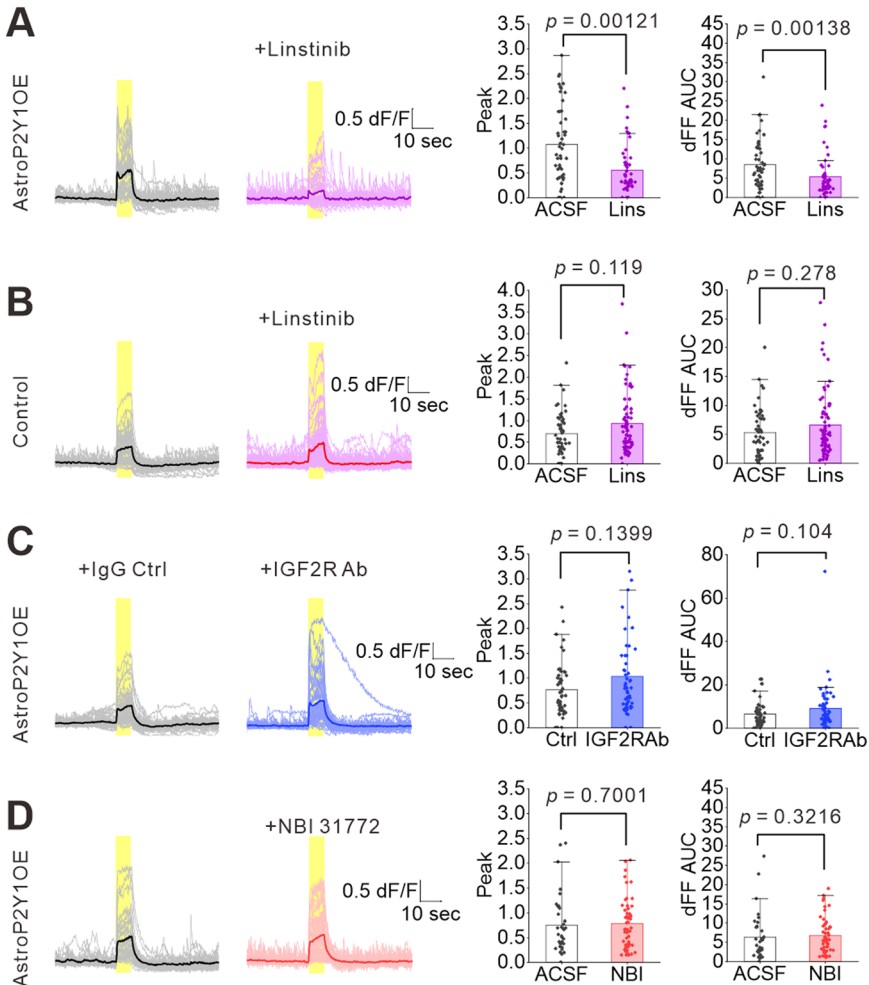

**Fig. 6 | Effects of pharmacological IGFBP2 signaling blockade on neuronal Ca²⁺ signals.** Stimulus-evoked neuronal Ca²⁺ signals in AstroP2Y1OE or control mice, measured using jGCaMP8s. The Schaffer collaterals were stimulated 300 times with 40 Hz at 0.06 mA. **A** Linsitinib (10 µM), an IGF-1R antagonist, was treated for 1 h at room temperature. Linsitinib significantly reduced neuronal Ca²⁺ signal in AstroP2Y1OE slices. ACSF, n = 51 ROIs from 4 slices, 3 mice; Linsitinib, n = 52 ROIs from 4 slices, 3 mice (two-sided Mann–Whitney U test). Data are presented as mean ± s.d. **B** Linstinib did not affect neuronal Ca²⁺ signals in control slices. ACSF, n = 53 ROIs from 6 slices, 4 mice; Linstinib, n = 71 ROIs from 6 slices, 4 mice (two-sided

Mann–Whitney U test). Data are presented as mean ± s.d. **C** IGF-2R antibody (5 µg/mL) was treated for 2 h at room temperature. IgG was used as a control. IgG control, n = 50 ROIs from 5 slices, 3 mice; IGF-2R antibody, n = 50 ROIs from 5 slices, 3 mice (two-sided Mann–Whitney U test). Data are presented as mean ± s.d. **D** NBI 31772 (10 µM), which blocks IGFBP binding to IGF-1, was treated for 1 h at room temperature. ACSF, n = 35 ROIs from 5 slices, 4 mice; NBI 31772, n = 52 ROIs from 6 slices, 4 mice (two-sided Mann–Whitney U test). Data are presented as mean ± s.d. Source data are provided as a Source Data file.

other neurological disease models. However, we did not find obvious upregulation of these genes in astrocytes in those datasets (Table 1).

## Discussion

Reactive astrocytes play a pivotal role in the pathogenesis of neurological diseases. Although several types of reactive astrocytes have been characterized at the molecular level, their functional phenotypes remain largely unknown[14,15]. In the present study, we demonstrated one of the functional phenotypes of reactive astrocytes characterized by the upregulation of P2Y1R, a Gq-GPCR, which leads to robust and large Ca²⁺ signals and upregulates IGFBP2 in astrocytes. Our data indicate that P2Y1R upregulation causes neuronal hyperexcitability by enhancing both neuron–astrocyte and astrocyte–neuron transmission without altering the "reactivity" of astrocytes, at least at the transcriptome level. We also revealed that IGFBP2 acts as an excitatory signal derived from astrocytes under the control of P2Y1R signaling. Furthermore, the finding that reactive astrocytes that express high levels of P2Y1R have upregulated IGFBP2 in epilepsy and stroke models suggests that P2Y1R-IGFBP2 signaling may be relevant to their pathogenesis, likely accompanied by neuronal hyperexcitability (Fig. 8).

Astrocyte-specific P2Y1R overexpression increased seizure susceptibility and abnormal spikes in the current study, indicating that an increase in P2Y1R signaling can cause neuronal hyperexcitability. These data support the view that P2Y1R activation is required for neuronal hyperexcitability in several types of neurological disease models[26,28,30,64,65]. Several downstream molecules of P2Y1R, including glutamate release[28], pannexins[30], and thrombospondin I[65], have been proposed to induce neuronal excitation. Of these, many studies have suggested that Ca²⁺-dependent glutamate release causes neuronal excitation via P2Y1R activation in astrocytes[27,48–50]. However, our extracellular glutamate imaging findings suggest that P2Y1R activation does not cause obvious glutamate elevations but does lead to massive Ca²⁺ elevation in astrocytes in AstroP2Y1OE mice (Fig. 3). Moreover, our RNA-seq data revealed no changes in genes related to glutamate homeostasis in astrocytes when P2Y1R was overexpressed in astrocytes (Fig. 4H). Instead, we found that IGFBP2 was upregulated by P2Y1R overexpression, which enhanced neuronal excitation. Pharmacological blockade of P2Y1R abolished Ca²⁺ signals in astrocytes in response to neuronal activity but did not affect neuronal Ca²⁺ signals in AstroP2Y1OE mice (Fig. 2D, E). These results suggest that P2Y1R activation in

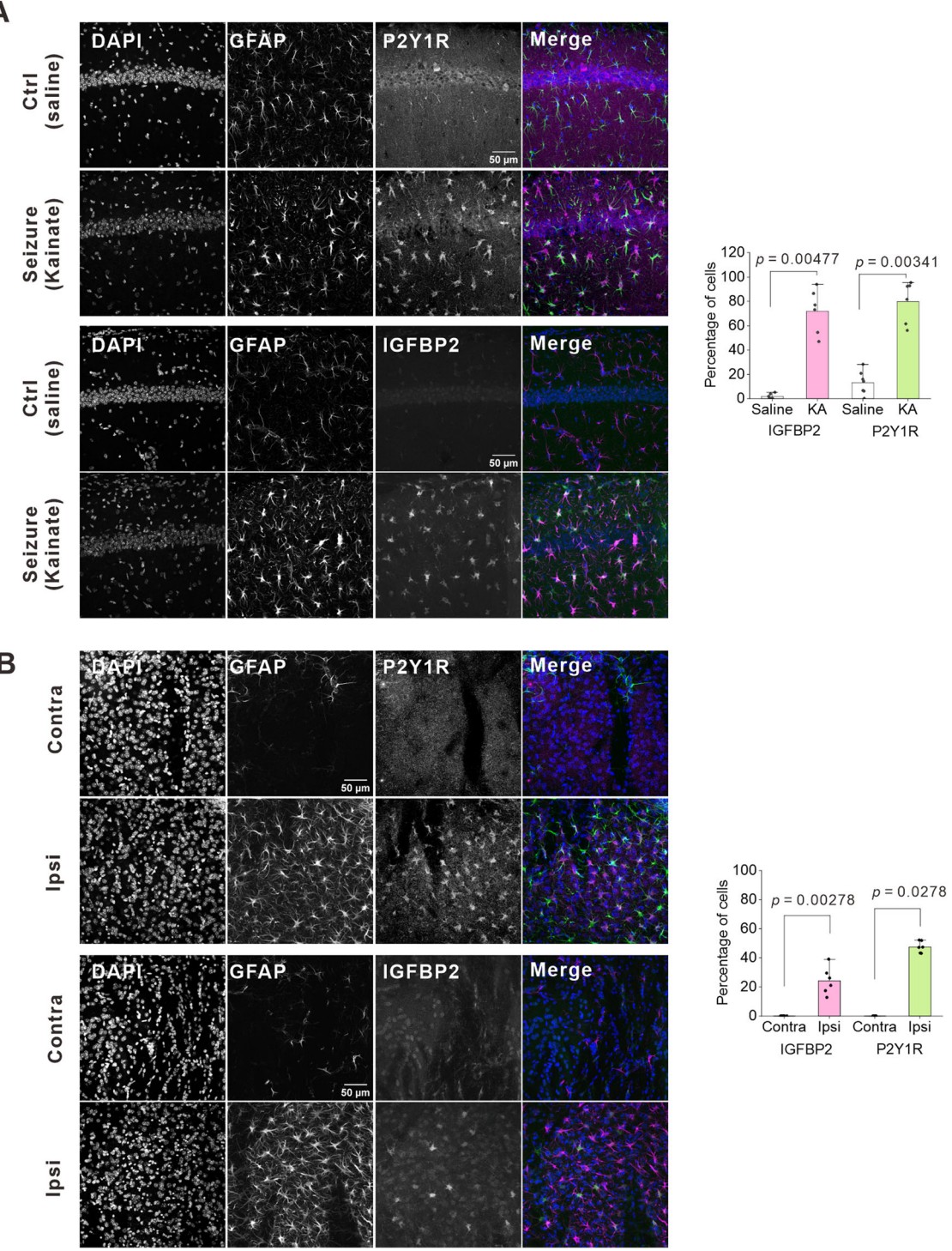

**Fig. 7 | IGFBP2 and P2Y1R are co-upregulated in reactive astrocytes in mouse models of seizure and stroke. A** Representative images of GFAP, P2Y1R, and IGFBP2 immunohistochemistry in the hippocampal CA1 region. The images were taken from the side contralateral to the kainic acid injection. Note that P2Y1R and IGFBP2 were upregulated in reactive astrocytes. These astrocytes were similarly distributed in the hippocampal CA1 region. The bar graph shows a summary of P2Y1R- and IGFBP2-positive GFAP-positive astrocytes. n = 6–7 slices from 3 mice for each group (two-sided Mann–Whitney U test). Data are presented as mean ± s.d.

**B** Immunohistochemistry for IGFBP2 and P2Y1R in astrocytes was performed in brains 3 days after MCAO. Expression of IGFBP2 and P2Y1R appeared in GFAP-positive astrocytes in the striatum ipsilateral to MCAO. There was no obvious expression of either P2Y1R or IGFBP2 in the side contralateral to MCAO. The bar graph shows a summary of P2Y1R- and IGFBP2-positive GFAP-positive astrocytes. n = 6 slices from 4 mice (two-sided Mann–Whitney U test). Data are presented as mean ± s.d. Source data are provided as a Source Data file.

AstroP2Y1OE mice upregulates *Igfbp2* and subsequently IGFBP2 protein but does not induce IGFBP2 secretion immediately after P2Y1R activation. Our data indicate that glutamate release is facilitated at CA3–CA1 synapses in response to repetitive stimulation in AstroP2Y1OE

mice, thus suggesting a presynaptic effect (Figs. 1D, E and 3, Supplementary Fig. 2I–K). However, the effect of IGFBP2 on neuronal excitation seemed to be larger at the soma (where relatively few excitatory synapses exist) than in dendrites (Fig. 5D, F, Supplementary

**Table 1 | Astrocytic expression of *P2ry1/P2RY1* and *Igfbp2/IGFBP2* in neurological disease models in publicly available datasets**

| GEO accession | Organism | Disease model | Experiment type | Comparison | Gene expression | | | | |
|---|---|---|---|---|---|---|---|---|---|
| | | | | | *P2ry1/P2RY1* | | *Igfbp2/IGFBP2* | | |
| | | | | | Log2FoldChange | FDR | Log2FoldChange | FDR | |
| GSE183050 | Mus muscles | Alzheimer's disease model | RNA-seq | APPPS1::APOE4fl/fl::ALDH1L1-CreERT vs APPPS1::APOE4fl/fl | −0.28 | 0.99 | −0.47 | 0.96 | |
| GSE183050 | Mus muscles | Alzheimer's disease model | RNA-seq | APOE4fl/fl::ALDH1L1-CreERT vs APPPS1::APOE4fl/fl::ALDH1L1-CreERT | 0.47 | 0.95 | 0.80 | 0.90 | |
| GSE129797 | Mus muscles | Alzheimer's disease model | RNA-seq | WT vs Tau-P301S | −0.30 | 0.76 | −2.4 | 0.18 | |
| GSE138695 | Homo sapience | Alzheimer's disease | RNA-seq | Ctrl vs AD4 | 0.70 | $2 \times 10^{-6}$ | 0.19 | 0.063 | |
| GSE138695 | Homo sapience | Alzheimer's disease | RNA-seq | Ctrl vs AD5 | 1.13 | $1 \times 10^{-14}$ | 0.13 | 0.22 | |
| GSE191131 | Mus muscles | Parkinson's disease model | RNA-seq | WT vs parkinsonian | 0.29 | 0.97 | −0.085 | 0.98 | |
| GSE103782 | Mus muscles | Stroke model | RNA-seq | Ctrl vs MCAO | 0.72 | 0.59 | −0.77 | 0.46 | |

Astrocyte-specific gene expression in several disease models from Publicly available datasets was analyzed with iDEP.96 and/or OlvTools.

Figs. 6 and 7D). Exogenous IGFBP2 increases firing and enhances excitatory synaptic transmission at the same synapses via a postsynaptic effect during development[57]. Astrocyte-derived IGFBP2 therefore probably acts via IGF-1R on both presynaptic and postsynaptic sites in neurons—to regulate glutamatergic synaptic transmission and increase firing, respectively—thus leading to hyperexcitability in AstroP2Y1OE mice (Fig. 8).

Astrocytes respond to neuronal insults such as brain injury and infection, becoming reactive astrocytes. Reactive astrocytes are highly heterogeneous, recruiting multiple biological pathways with both upregulation and downregulation of many genes via combinatorial gene expression regulation[11,12,14,58,66,67]. These transcriptome changes are very divergent; it is therefore difficult to elucidate which gene expression may be critical for different astrocytic functions. In the present study, we focused on the P2Y1R-mediated signaling pathway, which is commonly altered in several neurological disorders such as epilepsy, stroke, AD, and traumatic brain injury[25-33]. We explored its pathophysiological significance by simply overexpressing P2Y1Rs in astrocytes, without any neuronal insults. Our results suggest that astrocyte-specific P2Y1R overexpression leads to neuronal hyperexcitability without any signs of reactive astrocytes at the transcriptome level (Fig. 4C), suggesting that P2Y1R may not be a driver for reactive astrocytes. Moreover, astrocytic P2Y1R overexpression did not alter the expression of genes related to homeostatic functions, such as ion/neurotransmitter homeostasis (Fig. 4H). Interestingly, the RNA-seq data led us to discover IGFBP2, an excitatory molecule that enhances neuronal excitation. The finding that astrocytic P2Y1R overexpression did not alter reactive astrocyte genes suggests that P2Y1R upregulation may not be directly related to the "reactivity" of astrocytes, but may rather be functionally related to reactive astrocytes. IGFBP2 is co-upregulated with P2Y1R in reactive astrocytes in seizure and stroke models. This observation suggests that IGFBP2 represents a functional phenotype of reactive astrocytes.

In disease models in which reactive astrocytes have upregulated P2Y1R, pharmacological approaches can effectively ameliorate the symptoms of AD[26], epilepsy[64,68], stroke[31,32], and traumatic brain injury[33]. For example, Reichenbach et al. reported that chronic treatment with MRS2179, a P2Y1R antagonist, using an osmotic pump ameliorates deficits in synapses, long-term potentiation, and spatial learning memory in APPPS1 mice, an AD model[26], indicating that P2Y1R may be a therapeutic target for AD. However, although the blockade of P2Y1R can ameliorate disease pathogenesis, the effect may be dependent on context. For example, Alves et al. demonstrated that P2Y1R blockade has a dual effect on seizures; blockade of P2Y1R before seizure induction is pro-convulsant, whereas blockade after seizure induction is anti-convulsant[64]. This finding implies that timing is likely important for precision treatment using a pharmacological approach. Furthermore, because pharmacological approaches are not cell-type specific, unwanted side effects may occur[17]. Indeed, inhibitory neurons[69,70], microglia[71], and endothelial cells[72] also express P2Y1R. Given that the activation of those receptors may be neuroprotective, P2Y1R blockade in these cell types might have a detrimental effect on disease. To avoid these potential problems using pharmacological approaches, a specific downstream molecule of astrocytic P2Y1R is needed. To this end, we identified IGFBP2 in astrocytes of adult mice. Five lines of evidence in our study suggest that IGFBP2 acts as an excitatory signal under the control of P2Y1R in astrocytes of adult animals. First, IGFBP2 was expressed in astrocytes and its expression was higher in AstroP2Y1OE mice than in control mice. This finding is consistent with recent single-cell data showing that IGFBP2 is a marker of gray matter in the cortex[12,54]. Second, the inhibition of IGFBP2 signals reduced neuronal excitation induced by Schaffer-collateral stimulation. Third, exogenous IGFBP2 application enhanced neuronal excitation. Fourth, the knockdown of astrocytic *Igfbp2* reduced neuronal excitation. Fifth,

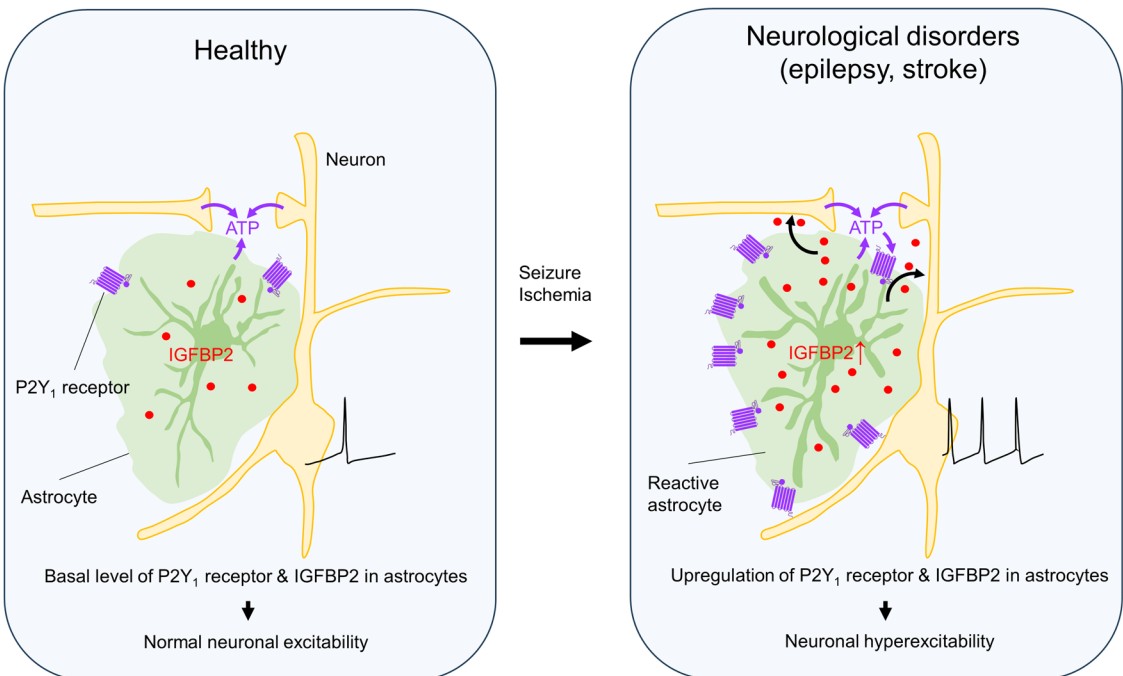

**Fig. 8 | Summary of the main findings.** P2Y1R is upregulated in reactive astrocytes in several neurological disorders such as epilepsy, stroke, Alzheimer's disease, and traumatic brain injury. The increase in P2Y1R signaling in astrocytes leads to neuronal hyperexcitability by increased expression of IGFBP2, which acts as an excitatory signal on neurons. Our data indicate that the functional phenotype of reactive astrocytes with upregulated P2Y1R-IGFBP2 signaling is shared in several neurological diseases including epilepsy and stroke and may contribute to neuronal hyperexcitability.

P2Y1R and IGFBP2 were upregulated in models of both seizure and stroke.

The mechanism underlying IGFBP2 upregulation via P2Y1R remains to be elucidated. One possibility involves $Ca^{2+}$-dependent transcription regulation through P2Y1R activation[73,74]. In support of this idea, recent systematic RNA-seq analysis of striatal astrocytes revealed that reduced $Ca^{2+}$ signals in astrocytes lead to reduced *Igfbp2* expression, suggesting that $Ca^{2+}$ signals regulate astrocytic *Igfbp2* expression, at least in the striatum[75]. The reduction of astrocytic $Ca^{2+}$ signals by CalEx leads to an approximately two-fold reduction of *Igfbp2* expression. Moreover, neuronal ablation by 1-methyl-4-phenyl-1,2,3,6-tetrahydropyridine causes an approximately four-fold increase in *Igfbp2* expression, suggesting that neuronal damage and subsequent ATP release may be a trigger for its expression. In both seizure and stroke models, neurons undergo damage as a result of excitotoxicity or ischemic insult. This neuronal damage may therefore be a trigger of IGFBP2 upregulation in astrocytes. However, the $G_q$-designer receptor exclusively activated by designer drugs (DREADD) activation did not affect IGFBP2 expression. This seems somewhat controversial but may suggest that a transient increase in $Ca^{2+}$ is not enough to induce IGFBP2 expression. Rather, the sustained activation of $G_q$-GPCR signaling may be required for IGFBP2 induction. Although we have not yet revealed the detailed mechanism underlying IGFBP2 upregulation, our aforementioned RNA-seq data suggest that P2Y1R activation triggers IGFBP2 expression via $Ca^{2+}$ signals.

Although IGFBP2 can be released from astrocytes upon any kind of brain insult[76], the mechanism for IGFBP2 release from astrocytes remains largely unknown. Activation of P2Y1R may trigger the release of IGFBP2 from astrocytes. However, given that IGFBP2 is a downstream molecule of P2Y1R, brief antagonism of P2Y1R inhibited astrocytic but not neuronal $Ca^{2+}$ signals in neurons of AstroP2Y1OE mice (Fig. 2D, E), suggesting that IGFBP2 may be released in a $Ca^{2+}$-independent manner. Thus, P2Y1R-mediated $Ca^{2+}$ signaling may not regulate IGFBP2 release, but rather the induction of *Igfbp2* expression in astrocytes.

AstroP2Y1OE increases the firing of neurons, the probability of glutamate release, evoked glutamate level, and neuronal $Ca^{2+}$ signals suggesting that IGFBP2 could act on both presynaptic and postsynaptic sites of neurons. Several mechanisms of IGFBP2 action have been proposed; these include IGF-1, IGF-2, IGF-1-R, and IGF-2R[61,76]. Exogenous IGFBP2 application enhances neuronal excitation by increasing excitatory synaptic transmission, firing, and long-term potentiation in the hippocampus in development[57], which is mediated by IGF-1R. The systemic application of IGFBP2 increases mature spine number in the medial prefrontal cortex and hippocampus[77]. A follow-up study revealed that a small peptide of IGFBP2 increases spine number via IGF-2R[61]. However, our pharmacological data suggest that IGF-1R but not IGF-2R contributes to IGFBP2 signals. Given that both neurons and astrocytes express IGF-1R[78,79], the cell types that contribute to neuronal excitation remain elusive; however, it has been reported that presynaptic IGF-1R positively regulates excitatory synaptic transmission in the CA1 region of the hippocampus[80]. Although PPR data in Supplementary Fig. 2I−K suggest an increase in the probability of glutamate release, we did not find a change in the input-output curve (Supplementary Fig. 2F−H), which may not strongly support the idea of the increase in the release probability. Another possibility is that IGFBP2 inhibits IGF signaling by sequestering IGF from its receptors[58], which may lead to increased extracellular glutamate levels via reduced IGF-1 signaling in astrocytes[79]. Given that IGFBP2 alters the structure of synapses[61], astrocytic coverage of synapses may change to modulate glutamatergic transmission because the proximity of astrocyte leaflets affects glutamate diffusion and spillover[81]. Those changes could also contribute to the enhancement of glutamatergic transmission by IGFBP2. Future studies are needed to reveal the mechanisms of action of IGFBP2 on glutamatergic transmission and firing.

The current study revealed a close link between P2Y1R and IGFBP2 in astrocytes. In models of both seizure and stroke, reactive astrocytes upregulated both P2Y1R and IGFBP2; this may be a common functional phenotype of reactive astrocytes in both disease models. In addition to

seizure and stroke, this phenotype of reactive astrocytes might be relevant to AD. A series of papers from Petzold et al. indicate that astrocytic P2Y1R upregulation correlates with neuronal hyperexcitability and exacerbates synaptic and memory deficits in AD model mice[25,26]. Furthermore, in a different AD model, IGFBP2 was upregulated in aged (18-month-old) AD model mice[82]. Although further investigation is required, these reports suggest that astrocytes with upregulated P2Y1R exert their function via IGFBP2 production, thus contributing to AD pathogenesis. Analysis of publicly available databases of Alzheimer's disease models (GSE183050, GSE129797, GSE138695) did not reveal changes in *Igfbp2/IGFBP2*. The absence of upregulation of the genes is unknown but may be due to differences in age, strain, severity of the disease models, and variability in biological replicates.

In the present study, immunohistochemical data revealed that IGFBP2- and P2Y1R-positive astrocytes were similarly distributed in tissue affected by epilepsy (CA1) and stroke (dorsal striatum). GFAP was also highly upregulated in the regions where IGFBP2 and P2Y1R were high. Given that IGFBP2 enhances excitatory synaptic transmission, as shown in past studies[57,58] and the present study, IGFBP2 may contribute to hyperexcitability and/or excitotoxicity in these diseases. In the current study, we focused on the early stages of disease models. It may be that the role of IGFBP2 is dependent on the disease stage. For example, the RNA-seq analysis of tissue from patients with mesial temporal lobe epilepsy-hippocampal sclerosis revealed markedly reduced *Igfbp2* expression[83]. To better understand the relationship between disease trajectory and P2Y1R-IGFBP2 signaling, the spatiotemporal expression patterns of P2Y1R and IGFBP2 in astrocytes need to be investigated.

In summary, our results indicate that astrocytes with upregulated P2Y1R trigger the expression of IGFBP2, which works as an astrocyte-derived secretory signaling molecule that leads to neuronal hyperexcitability. There may be a common link between P2Y1R and IGFBP2 in reactive astrocytes in several neurological diseases, such as epilepsy and stroke. Thus, IGFBP2 may be a therapeutic target for neurological diseases in which reactive astrocytes upregulate P2Y1R.

In our model, P2Y1Rs were overexpressed throughout all stages of development. We therefore cannot exclude the possibility that P2Y1R overexpression in astrocytes alters neuronal circuits and astrocytic networks in development. However, both the genetic and pharmacological inhibition of IGFBP2 signals reduced neuronal Ca$^{2+}$ signals, indicating that astrocyte-derived IGFBP2 regulates neuronal Ca$^{2+}$ signals in brain slices from adult mice. Another limitation of the current study is that the AstroP2Y1OE mice had lower body weights (Supplementary Fig. 9), smaller hippocampi, and enlarged ventricles compared with control mice. These phenotypes may be partly caused by increased IGFBP2 expression because IGFBP2 overexpression reduces postnatal weight gain[84]. However, these changes may also have affected the behavioral phenotypes that were observed. Nevertheless, the finding of both IGFBP2 and P2Y1R upregulation in the astrocytes of two distinct disease models, and the data indicating that IGFBP2 enhances neuronal excitation, indicate that our identified mechanism is relevant to astrocyte–neuron communication in neurological diseases.

## Methods
### Animals
All procedures were performed in accordance with the "Guiding Principles for the Care and Use of Animals in the Field of Physiologic Sciences" published by the Physiologic Society of Japan and with the previous approval of the Animal Care Committee of Yamanashi University (Chuo, Yamanashi, Japan; Approval No. A29-7). To regulate P2Y1R expression specifically in astrocytes, we took a transgenic approach using the Tet-Off system. Two transgenic lines were crossed: the Mlc1-tTA BAC transgenic line and the P2ry1tetO knockin line[36,37,85]. Double transgenic mice are referred to as AstroP2Y1OE mice.

Doxycycline is not required to drive P2Y1R overexpression. In all the experiments except Fig. 6, littermate P2ry1tetO knockin line mice were used as control. Mice were housed on a 12-h light (6 a.m.)/dark (6 p.m.) cycle with ad libitum access to water and rodent chow. Male mice were used. Genotyping polymerase chain reaction (PCR) was performed using DNA purified from tail biopsies. The PCR primers used were: P2ry1tetO forward, 5′-gtaggcgtgtacggtggag-3′; P2ry1tetO reverse, 5′-tgcagttgaggcgacagtac-3′; Mlc1-tTA forward, 5′-aaattcaggaagctgtgtgcc tgc-3′; Mlc1-tTA reverse, 5′-cggagttgatcaccttggacttgt-3′. Male C57BL6/J mice were purchased from Japan SLC and used for the experiments on kainite-induced seizures and MCAO.

### Recombinant AAV vector production
Synthetic oligonucleotides included the targeting sequence for exon 2 of *Igfbp2* (5′-gaacagcaccggcagatgggc-3′) with the targeting site in a pENTER-U6-sgBsa1 plasmid. The guide sequence was designed using CRISPOR to minimize potential off-target effects[86]. The resulting U6-sgRNA cassette was transferred into pZac2.1-GfaABC1D-mCherry[59]. pZac2.1-GfaABC1D-SaCas9 and pZac2.1-GfaABC1D-mCherry plasmids were prepared. All the synthetic oligonucleotides and sequencing primers were obtained from Eurofins Genomics. Recombinant AAV vectors were produced from HEK293 cells (Agilent) with triple transfection (pZac2.1-GfaABC1D.Igfbp2-single-guide RNA mCherry plasmid, pAAV2/5 [Addgene Cat# 104964], and pAdDeltaF6 [Addgene Cat# 112867]). Viral lysates were harvested 72 h after transfection and lysed using freeze–thaw cycles, purified through two rounds of CsCl ultracentrifugation, and concentrated with Vivaspin 20 ultrafiltration units (SARSTEDT). The genomic titer of recombinant AAV was determined using PicoGreen fluorometric reagent (Molecular Probes) after denaturation of the AAV particle. Vectors were stored in aliquots at −80 °C until use.

### Immunohistochemistry
Mice were anesthetized with an anesthetic mixture of medetomidine (0.75 mg/kg, Kyoritsu Seiyaku), midazolam (4 mg/kg, Maruishi Pharmaceutical), and butorphanol (5 mg/kg, Meiji Animal Health). Mouse brains were fixed with 4% paraformaldehyde (PFA) via cardiac perfusion. After 12–14 h of post-fixation in 4% PFA, brains were submerged in 30% sucrose in phosphate-buffered saline (PBS) for 2–4 days. Frozen 20 (Fig. 5A and Supplementary Fig. 5C, D), 40 μm (Fig. 7A and Supplementary Figs. 5A, B and 7A, B), or 50 μm (Fig. 7B and Supplementary Fig. 8) sections were cut using a cryostat (Leica Microsystems). Slices were washed three times with PBS before being treated with 0.5% Triton-X100/10% normal goat serum (VectorLaboratories Cat# S-1000-20) for 1 h. After three washes with PBS, sections were incubated for 48 h with the following primary antibodies: rabbit anti-IGFBP2 antibody (1:500, Abcam, RRID:AB_2938998), mouse anti-IGFBP2 antibody (1:100, Santa Cruz Biotechnology, Cat# sc-515134), rat anti-GFAP antibody (1:2000, Thermo Fisher Scientific; RRID: AB_2532994), rabbit anti-P2Y1R antibody (1:250, Alomone Labs; RRID: AB_2040070), rabbit anti-SOX9 antibody (1:1000, Millipore; RRID: AB_2239761), rat anti-mCherry (16D7) (1:2000, Thermo Fisher Scientific, RRID: AB_2536611), mouse anti-hemagglutinin (HA) antibody (1:500, BioLegend, RRID: AB_2565336), guinea pig anti-GLT-1 antibody (1:1000, Sigma-Aldrich, RRID:AB_90949), and rabbit anti-GLAST antibody (1:200, Nittobo Medical, MSFR102120). Sections were then washed three times with PBS before being incubated for 2 h with the following fluorescent secondary antibodies (all 1:500, Thermo Fisher Scientific): anti-mouse IgG Alexa Fluor 488 (RRID: AB_2536161), anti-rabbit IgG Alexa Fluor 405 (RRID: AB_221605), anti-rabbit IgG Alexa Fluor 488 (RRID: AB_143165), anti-rabbit IgG Alexa Fluor 546 (RRID: AB_2534093), anti-rat IgG Alexa Fluor 488 (RRID: AB_2534125), anti-guinea pig IgG Alexa Fluor 488 (RID:AB_2534117), and anti-rat IgG Alexa Fluor 546 (RRID: AB_2534125). After washing three times with PBS, sections were mounted on slides. Images were captured using a Fluoview 1200 (Olympus) equipped with

a 40× silicon oil immersion (numerical aperture 1.25) or 40× dry lens (numerical aperture 0.95).

## Microinjections of AAV

Male mice aged 7–9 weeks were anesthetized with isoflurane (induction at 2%–3%, maintenance at 1.0%–1.5% volume). The depth of anesthesia was monitored continuously and adjusted when necessary to ensure that the animals were always anesthetized. Following anesthesia induction, mice were fitted into a stereotaxic frame with their heads secured by blunt ear bars and their noses placed into an anesthesia and ventilation system. The mice were administered 0.1 mL carprofen (Rimadyl®, 1.5 mg/mL) subcutaneously before surgery for pain relief. The surgical incision site was cleaned three times with povidone–iodine and 70% ethanol, and the skin was treated with 2% Xylocaine® Jelly for pain relief. Skin incisions were made, followed by periosteum removal after the application of 1% lidocaine (Xylocaine, 10 mg/mL). Craniotomies of approximately 0.5–2 mm in diameter were made above the parietal cortex using a small (0.4 mm) steel burr (Minitor) powered by a high-speed drill (Minimo, Minitor). Artificial cerebrospinal fluid (ACSF), composed of (in mM) 152 NaCl, 2.5 KCl, 1 $MgCl_2$, 2 $CaCl_2$, 10 HEPES, pH 7.3, approximately 310 mOsm, was applied onto the skull to reduce heat caused by drilling. Unilateral or bilateral viral injections were performed using beveled glass pipettes (1B100-4, WPI; 35°, outer diameter, approximately 40 μm) driven by a syringe pump (UMP-3+micro4, WPI). For dual-color $Ca^{2+}$ imaging, 1500 nL of the mixture of AAV2/5 GfaABC1D.cyto-GCaMP6f.SV40 ($9 \times 10^{12}$ GC/mL, Addgene Cat# 52925) and AAV2/1 Syn.NES.jRGECO1a.WPRE.SV40 ($3.4 \times 10^{12}$ GC/mL, Addgene Cat# 100854) was injected unilaterally into the left hippocampus (2 mm posterior to the bregma, 1 or 1.5 mm lateral to the midline, approximately 1.3–1.5 mm from the pial surface) at a speed of 0.2 μL/min. For extracellular glutamate imaging, 1000 nL of AAV2/5. GfaABC1D.GluSnFr.SV40 ($2.0 \times 10^{13}$ GC/mL, Penn Vector Core), the first generation of iGluSnFR, was injected into the left hippocampus at a speed of 0.2 μL/min. For extracellular ATP imaging, 500 nL ($3.56 \times 10^{13}$ GC/mL) or 1000 nL ($1.78 \times 10^{13}$ GC/mL) of AAV9-gfaABC1D-ATP1.0 (WzBiosciences) was injected into the left and/or right hippocampus at a speed of 0.2 μL/ min. For IGFBP2 genome editing, 600 nL of the mixture of AAVs (AAV2/5.GfaABC1D.SaCas9 [$3.0 \times 10^{12}$ GC/mL] + AAV2/5.GfaABC1D.m-Cherry [$2.0 \times 10^{12}$ GC/mL] or AAV2/5.GfaABC1D.SaCas9 [$3.0 \times 10^{12}$ GC/ mL]+AAV2/5.GfABC1D.Igfbp2.sigle-guide RNA [$2.0 \times 10^{12}$ GC/mL]) was injected bilaterally into the hippocampus (2 mm posterior to the bregma, 1.5 mm lateral to the midline, approximately 1.3 mm from the pial surface) at a speed of 0.2 μL/min. For neuronal $Ca^{2+}$ imaging, 600 nL of AAV2/1 jGCaMP8s ($5.0 \times 10^{12}$ GC/mL, Addgene Cat# 162374) was injected bilaterally into the hippocampus (2 mm posterior to the bregma, 1.5 mm lateral to the midline, approximately 1.3 mm from the pial surface) at a speed of 0.2 μL/min. The AAVs were diluted to titers with 0.001% Pluronic F-68 (Thermo Fisher Scientific) in PBS. The glass pipettes were left in place for at least 10 min following each injection to minimize the backflow of the viral solution. Surgical wounds were closed with single external 5-0 nylon sutures (Akiyama Medical). Carprofen was administered once daily for up to 3 days after surgery. In addition, enrofloxacin (Baytril®, 85 mg per 1 L water) was dispensed in drinking water to prevent infections until the mice were used in experiments. Hippocampal slices (300 μm) were cut and imaged 3–4 weeks post-surgery.

## Preparation of brain slices, $Ca^{2+}$ imaging, and electrophysiology

We cut acute brain slices for $Ca^{2+}$ imaging and electrophysiological recordings. Briefly, male mice were anesthetized with isoflurane (approximately 2% volume/volume). Cold-cutting ACSF, composed of (in mM) 93 N-methyl-D-glucamine, 93 HCl, 2.5 KCl, 1.2 $NaH_2PO_4$, 30 $NaHCO_3$, 20 HEPES, 25 D-glucose, 5 sodium ascorbate, 2 thiourea, 3 sodium pyruvate, 10 $MgCl_2$, and 0.5 $CaCl_2$ saturated with 95% $O_2$–5%

$CO_2$, was perfused transcardially. Coronal slices of the hippocampus (300 μm) were cut on a vibrating microtome (Pro7, Dosaka, or VT1200S, Leica Microsystems) in cutting ACSF. We used slices between −1.6 and −2.3 mm from the bregma to reduce the variability of the responses. Slices were then incubated at 34 °C for 10 min in cutting ACSF and subsequently stored in recording ACSF, comprised of (in mM) 124 NaCl, 2.5 KCl, 1.2 $NaH_2PO_4$, 24 $NaHCO_3$, 5 HEPES, 12.5 D-glucose, 2 $MgCl_2$, and 2 $CaCl_2$ saturated with 95% $O_2$ and 5% $CO_2$ at room temperature. After 1 h of recovery, slices were submerged in recording ACSF at approximately 32 °C. Slices were then imaged using an Olympus Fluoview FVMPE-RS two-photon laser scanning microscope (Olympus) equipped with Maitai HP DS-OL and Insight DS-OL (Spectra-Physics). We used a 920 nm laser (7%–10% power) and a 495–540 nm bandpass emission filter for GCaMP6f, iGluSnFR, and jGCaMP8s and an 1100 nm laser (5%–10% of power) and a 575–645 nm bandpass emission filter for jRGECO1a and Alexa546. For some of the iGluSnFR experiments (Fig. 3A–D), a 510 nm high-pass filter was used. Images were captured from the stratum radiatum of the CA1 hippocampal region and were typically taken 50 μm from the slice surface. Images were observed using a 25× water immersion lens with a numerical aperture of 1.05 (Olympus). For dual-color $Ca^{2+}$ imaging, we took images 3 frames per second (pixel unit = 0.53 μm). For iGluSnFR imaging, we took images 5 frames per second (pixel unit = 0.53 μm). For jGCaMP8s imaging, we took images 5 frames per second (pixel unit = 0.53 μm). For GRABATP1.0 imaging, we took images every 0.3 s (pixel unit = 0.33 μm) For the EFS experiments, a concentric bipolar stimulus electrode (IMB-160820, Intermedical; interpolar distance, 10 μm) was placed on the Schaffer collaterals. To avoid movement artifact of the field of view (FOV) during EFS, the electrode was placed at -250–300 μm from the FOV. The EFS (pulse duration 100 μs, 40 Hz, 0.06 mA) was delivered by a stimulus isolator (IsoFlex, A.M.P.I) and controlled by a stimulator (Master9, A.M.P.I.). Although most drugs were applied in the bath, MRS2365 (10 μM, 5 psi, 500 ms) was applied locally using a puffer pipette (tip size, approximately 1 μm) connected to a Picopump (WPI) in the presence of TTX (0.5 μM). The MRS2365 puffer solution was prepared in recording ACSF with Alexa Fluor 568 hydrazide (50 μM) to visualize the drug application. Rabbit antibodies against IGFBP2 (5.91 μg/mL) or goat antibodies against IGF-2R (5 μg/ mL, Cat# AF2447) were added to the bath for 2 h, during which 95% $O_2$/ 5% $CO_2$ was continuously supplied. Control IgG (rabbit IgG isotype control, Cat# ab172730, or normal goat IgG, Cat# AB-108-C) was used as a control for the antibody treatment experiments. IGFBP2 (10 ng/ mL, BioLegend Cat# 750302), linsitinib (10 μM, Selleck Chemicals, Cat# S1091), or NBI 31772 (10 μM, Tocris Bioscience, Cat# 5192/10) were added to the bath for 1 h, during which 95% $O_2$/5% $CO_2$ was continuously supplied. After incubation, the slices were perfused with the reagent during the imaging experiments.

Electrophysiological recordings were performed at room temperature in recording ACSF saturated with 95% $O_2$/5% $CO_2$. Whole-cell patch-clamp recordings were made from granule neurons with a borosilicate glass pipette (WPI, Cat#1B150-4) using a MultiClamp 700B controlled by a computer running pCLAMP10.3 or pCLAMP10.5 software using a Digidata1440A interface (Axon Instruments). Data were filtered at 2 kHz and digitized at 5 kHz. The pipette solution for pyramidal neurons is comprised of 120 K-gluconate, 6 KCl, 2 $MgCl_2$, 5 $CaCl_2$, 10 EGTA, 2 ATPMg, 0.3 GTPLi, 10 HEPES (in mM), pH 7.2. The pipette solution for granule neurons in the dentate gyrus is comprised of 125 K-gluconate, 10 KCl, 4 ATPMg 10 EGTA, 10 Phosphocreatine2Na, 0.5 GTP2Na, 10 HEPES (in mM), pH 7.2. The resistance of the pipettes was 4–6 MΩ. Cells were visualized with infrared optics on an upright microscope (BX51Wl, Olympus). For voltage-clamp recording, the membrane potential was held at −60 mV. Cells showing the holding currents lower than −100 pA were omitted. Field EPSPs were recorded using an extracellular glass microelectrode (filled with recording ACSF; resistance = 5–8 MΩ placed in the striatum radiatum). The Schaffer-collateral fibers were stimulated

once every s using a concentric bipolar stimulus electrode (IMB-160820, Intermedical). The data were analyzed using Clampfit10.5 software (Axon Instruments).

## EEG recording
Mice were deeply anesthetized with isoflurane (2%). A bipolar electrode (Unique Medical) was implanted in the left CA1 area of the dorsal hippocampus (anterior/posterior = −1.8 mm, medial/lateral = +1.6 mm, dorsal/ventral = −2.0 mm). The electrode was fixed to the skull with dental cement (Estecem II, Tokuyama Dental), and mice were then allowed to recover for 5–7 days before EEG recording. EEGs were recorded in freely moving mice using a digital acquisition system (PowerLab 26T, ADInstruments). The EEG recordings were performed for at least 2 h, and data were collected at a sampling rate of 2000 Hz. At the end of the experiments, to make sure that we had successfully recorded EEG, we injected pilocarpine to induce seizures (Supplementary Fig. 1). Data were acquired, digitized, and analyzed offline using LabChart 8 software (ADInstruments). Any artifacts in the raw EEG traces were manually identified and excluded from the analyses of interictal spikes.

## Pilocarpine-induced status epilepticus
A low dose of 100 mg kg$^{-1}$ pilocarpine (Wako, 161-07201) per injection was administered intraperitoneally every 20 min until the onset of Racine scale stage 5 seizures[38]. Scopolamine methyl bromide (1 mg kg$^{-1}$, intraperitoneal, Wako, 198-07971) was administered 30 min before pilocarpine injection to reduce its peripheral effects.

## Kainate-induced seizures
Wild-type mice were anesthetized with isoflurane (1.5%–2.0%), and 70 (Fig. 7A, B) or 60 (Supplementary Fig. 7B) nL kainate solution (20 mM) was injected stereotaxically into the right cortex, above the right hippocampus (posterior to the bregma −1.9 mm, 1.5 mm lateral to the midline, 1.7 mm from the skull surface)[87] using a beveled pipette over a period of 1 min. After the injection, the pipette was left for 2 min to limit backflow. An equal volume of saline was injected for control animals. Four days after inducing status epilepticus, mice were fixed with 4% PFA for immunohistochemistry. We analyzed the contralateral side of the injection to avoid any effects of incision by the pipette.

## MCAO
Unilateral transient focal cerebral ischemia was induced in mice by intraluminal filament occlusion of the right middle cerebral artery. The animals were anesthetized with 6% (volume/volume) sevoflurane and maintained on 2.5% (volume/volume) sevoflurane with a face mask. Oxygen levels were maintained at 30% during the surgery. Rectal temperature was recorded and maintained at 36.0 °C–36.5 °C during the procedures using a homeothermic blanket. After a neck incision was made, the common carotid artery, internal carotid artery, and external carotid artery were exposed by dissection. Subsequently, the external carotid artery was ligated, and a silicone-coated monofilament suture (602556PK10Re, diameter 0.25 ± 0.02 mm, Doccol Corp) was inserted via the proximal external carotid artery into the internal carotid artery and then into the circle of Willis, thereby occluding the middle cerebral artery (MCA). The MCA was occluded for 15 min, and the suture was then carefully withdrawn to allow reperfusion of the ischemic region. Mice awakened spontaneously from anesthesia and were allowed free access to water and food. Three days after MCAO, the mice were fixed with 4% PFA. Their brains were removed, post-fixed in 4% PFA for 24 h, and sectioned on a vibratome (VT1200; Leica Microsystems) into coronal slices (40 μm).

## RNA-seq
Mice were anesthetized with isoflurane (Viatris) and then decapitated. Both hippocampi were dissected out. RNA-seq was performed using RNA extracted from isolated astrocytes from the hippocampi of male mice. To isolate astrocytes, we used gentleMACS dissociators and the Adult Brain Dissociation Kit (Miltenyi Biotec) according to the manufacturer's protocol. Briefly, bilateral hippocampi were dissected from three deeply anesthetized male mice (11 weeks old). Three mice were used for each sample. The tissue was minced and treated with an enzyme mix for 20 min at 37 °C. After enzymatic digestion, the tissue was gently triturated using a fire-polished Pasteur pipette, and debris was removed with a debris removal kit. To remove myeloid cells/microglia and oligodendrocytes, the cells were incubated at 4 °C with both anti-CD11b-coated beads and myeline beads II (Miltenyi Biotec) for 10 min. After washing the cell suspension to remove any unbound beads, the cell suspension was loaded onto an LS Column (Miltenyi Biotec) and placed into the magnetic field of a QuadroMACS™ Separator (Miltenyi Biotec) to collect the flowthrough. The cells were then incubated at 4 °C with anti-mouse astrocyte cell surface antigen-2 (ACSA2)-coated microbeads (Miltenyi Biotec) for 10 min. Next, the cell–bead mix was washed to remove unbound beads. To avoid non-specific binding of the ACSA2 antibody to the Fc receptor, cells were treated with FcR Blocking Reagent (Miltenyi Biotec) before incubation with the ACSA2 antibody. After washing to remove unbound beads, the cell suspension was loaded onto an LS Column (Miltenyi Biotec), which was then placed in the magnetic field of a QuadroMACS™ Separator (Miltenyi Biotec). The magnetically labeled ACSA2-positive cells were retained within the column and were eluted as the positively selected cell fraction after removing the column from the magnet. Next, RNA from the cells was extracted using an RNeasy Lipid Tissue Mini Kit (Qiagen) and stored at −80 °C. The RNA sample was sent to BGI Japan for RNA-seq analysis. Total RNA was amplified using Smart and subsequently sequenced using DNBseq PE100 (MGI). DNBSEQ Low Input Eukaryotic Transcriptome library construction procedure is as follows. (1) Take an appropriate amount of total RNA samples, and add oligo-dT reverse transcription primer, and denature the total RNA sample by heat, (2) Add the reverse mix reagent, and reverse transcribed to first-strand cDNA by SMART amplification technology, (3) Synthesize the second-strand cDNA, and use magnetic beads to purify the cDNA, and validate the cDNA, (4) The qualified double stranded cDNA, was used to construct the library with transposase, (5) The library was qualified by the Agilent Technologies 2100 bioanalyzer, (6) The library was circularized, (7) The library was amplified to make DNA nanoball (DNB) and sequenced on DNBSEQ platform. DEGs were analyzed using Dr. Tom software (BGI). Only RNA samples with RNA integrity number (RIN) > 8 were used for the analysis.

## Data analysis
We performed Ca$^{2+}$ imaging experiments on a single optical plane. Ca$^{2+}$ transients were measured by plotting the intensity of ROIs over time after the background intensity had been subtracted. The background intensity was calculated from regions without GECI expression. ROIs were selected from four regions of the stratum radiatum. Using these ROIs, raw fluorescence intensity values (F) were taken from the original images and converted to delta F/F (dF/F) in Origin 2022 (Origin Lab Corp) and Microsoft Excel. A signal was declared a Ca$^{2+}$ transient if it exceeded the baseline by five times that of the baseline standard deviation. Data were analyzed using Origin 2022 and FIJI[88]. Publicly available RNA-seq data were analyzed using iDEP.96 (http://bioinformatics.sdstate.edu/idep96/)[89] and/or OlvTools (https://olvtools.com/).

## Experimental design and statistical analysis
No statistical methods were used to pre-determine the sample size, but our sample sizes were consistent with those of similar studies. Fields of view that included morphologically unhealthy cells were excluded. Mice were randomly assigned to experiments. Data collection and analysis were not performed by experimenters blinded to the experimental conditions. The numbers of experimental replicates are

described in the figure legends. Statistical comparisons were made using the Mann–Whitney U test, unpaired Student's t-test, paired Student's t-test, one-way analysis of variance (ANOVA), or two-way repeated measure ANOVA after analyzing the raw data to check whether they were normally distributed. Data are shown as the mean ± standard error of the mean (Figs. 1E, F and 2A–C, Supplementary Figs. 2C, F–H, 3C, and 9) or standard deviation (others). Box-plot elements are defined in the following way: center line, median; box limits, upper and lower quartiles; whiskers, 1.5x interquartile range; square, mean. Statistical tests were performed in Origin 2022-2023.

### Reporting summary

Further information on research design is available in the Nature Portfolio Reporting Summary linked to this article.

## Data availability

All data needed to evaluate the conclusions in the paper are present in the paper and/or the Supplementary Materials. RNA-seq data generated in this study have been deposited in the GEO under accession code GSE242450. Source Data are provided with this paper. The publicly available data we analyzed in this study are as follows: GSE183050, GSE129797, GSE138695, GSE191131, GSE103782.

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

## Acknowledgements

This work was supported by JSPS Grants-in-Aid for Scientific Research (KAKENHI) grant numbers: 18H05121 (S.K.), 15KK0340 (E.S.), 17K01974 (E.S.), 21K06391 (E.S.), 20H05902 (S.K.), 21K19309 (S.K.), 23K18162 (S. K.), 24H00583 (S. K.) and 21H04786 (S.K.); JSPS KAKENHI grant number JP16H06276 (AdAMS); AMED-CREST grant number JP20gm131000 (S.K.); AMED-ASPIRE 23jf0126004h00 (S. K.); COI-NEXT (S. K.); the Kato Memorial Bioscience Foundation (E.S.); the Takeda Science Foundation (S.K. and E.S.); Suzuken Memorial Foundation (E.S.) and a GLIA Center Grant from the University of Yamanashi (S.K.). We thank Y. Fukasawa, M. Y. S. Yoshida, M. Shimizu, Y. Koseki, M. Tachibana, A. Kunugi, and R. Komatsu for their technical assistance. We thank all members of the Koizumi laboratory for their stimulating discussions. We also thank Bronwen Gardner, PhD and Bonnie Gardner, PhD, from Edanz (https://jp.edanz.com/ac) for editing a draft of this manuscript. Images in Figs. 1C and 5E were created using BioRender.com. We thank Drs. Zhaofa Wu and Yulong Li helped to obtain AAV9-GfaABC1D-ATP1.0. Transcriptomic analysis of publicly available datasets was performed using iDEP.96 (http://bioinformatics.sdstate.edu/idep96/) and/or another web application from OlvTools (https://olvtools.com/). Some of the images in Fig. 4A are from TogoTV (©2016 DBCLS TogoTV/CC-BY-4.0).

## Author contributions

E.S. and S.K. designed the research. E.S., Y.J.H., H.S., Y.N., and F.S. performed experiments and analyzed data. K.F.T., Y.S., and H.B. provided resources and critical feedback. E.S., K.Y., K.K., and M.T. designed and prepared AAV2/5 gfaABC1D.sgIgfbp2 and AAV2/5.GfaABC1D.Sa-Cas9. T.T., H.Y., and H.K. provided resources and helped with the immunohistochemistry experiments. E.S. wrote the manuscript. S.K. provided critical feedback. All authors contributed to the final version of the manuscript.

## Competing interests

The authors declare no competing interests.
