## [Peer Review File · Nature Communications]

REVIEWER COMMENTS

Reviewer #1 (Remarks to the Author):

The authors have tested the functional consequences of astrocytic P2Y1 receptor (P2Y1R) upregulation, which has been observed in several pathophysiological conditions. They used an impressive set of imaging, electrophysiological and molecular approaches to dissect the effects of P2Y1R overexpression. They found that P2Y1R overexpression increased seizure susceptibility and neuronal excitability in vivo and in vitro and also increased astrocytic Ca²⁺ signal activity in vitro. The authors further analysed the mechanisms underlying these observations in acute hippocampal slices and also performed glutamate imaging. They further characterised how P2Y1R overexpression affects RNA expression using RNA-seq and identified IGFBP2 as a potential link between P2Y1R overexpression and the observed functional changes. Interestingly, reducing IGFBP2 levels by neutralizing antibodies and reducing astrocytic IGFBP2 expression decreased neuronal activity whereas IGFBP2 incubation increased it. The authors also showed that the increased expression of P2Y1R and IGFBP2 is reproduced in animal models of epilepsy and stroke. These are important and novel findings. In contrast to many previous studies, which have described the many simultaneous changes of astrocyte physiology in disease and their potential implications, this study isolates the effect of one particular relevant change (expression of P2Y1R) and demonstrates how it can reproduce important aspects of disease pathophysiology. The use of P2Y1R overexpression model is elegant, although it does have some significant disadvantages probably due to the persistent P2Y1R upregulation as discussed by the authors.

Comments:

- 1) The authors dissect the activation of neurons and astrocytes in hippocampal slices after stimulation of Schaffer collateral synapses. They conclude that stimulus-induced astrocytic Ca²⁺ transients require action potential firing and are almost exclusively mediated by P2Y1R receptors. This seems at odds with reports from the same brain region (Tang et al. 2015, Panatier et al. 2011, among many others). What is the explanation for this discrepancy? It is stated that astrocytic Ca²⁺ signals are largely unaltered by CNQX/APV (line #186-188) although the reduction in Fig. 2A is highly significant. Based on further experiments the authors conclude that neuronal pannexin channels may release ATP. It is not clear how this may work. What is the trigger of pannexin-mediated ATP release? If it were due to postsynaptic depolarisation/activity it would be highly sensitive to CNQX/APV.
- 2) Experiments were done using iGluSnFR expressed by astrocytes. (Please state in the manuscript how and which iGluSnFR was expressed.) An increase in iGluSnFR signals was observed in P2Y1R-overexpressing animals. This could reflect an increase of glutamate release, a reduction in glutamate

uptake or a shifted balance between glutamate transporter and iGluSnFR expression (see for instance Armbruster et al. 2020). Can the authors distinguish these possibilities, also considering their findings from Suppl. Fig. 2F-h? The absence of mRNA changes of proteins relevant for glutamate homeostasis does not exclude changes of protein function/location/expression and, thereby, functional changes.

3) Why does IGFBP2 affect somatic but not dendritic Ca²⁺ signals of neurons (Fig. 5 and Suppl. Fig. 5) and 6)? If IGFBP2 does not significantly affect dendritic Ca²⁺ signals then it cannot explain the changes observed elsewhere (e.g. Fig. 1C-D), where presumably dendritic Ca²⁺ signals were included in the analysis and quantification since stratum radiatum was named as the region of interest for analysis.

4) The results in Fig. 6 are difficult to interpret in the context of P2Y₁R overexpression because there seems to be no comparison to control mice. If the effect of increasing IGFBP2 in P2Y₁R overexpressing animals is due to an interaction with the IGF1 receptor, then the effect of Linsitinib should be different between control and P2Y₁R overexpressing animals.

5) Why did the authors analyse the striatum for the MCAO experiment and not the hippocampus as done elsewhere? What happened in the cortex or hippocampus of the mice in these experiments?

6) The authors further speculate that IGFBP2 enhances glutamatergic synaptic transmission (line #358-360) but Suppl. Fig. 2F-H and the lack of a clear effect on dendritic Ca²⁺ signals argue against that.

7) The authors mention ,basal glutamate levels' in the discussion (line #347-349) but baseline or resting glutamate levels/concentrations have not been reported in the study.

Reviewer #2 (Remarks to the Author):

The study by Shigetomi et al. investigates the consequences of constitutive over-expression of P2Y₁ receptors in astrocytes on neuronal and astrocytic functions in the mouse hippocampus. P2Y₁ receptors are up-regulated in reactive astrocytes in different neurological disease (stroke, epilepsy, AD...) but the consequences of this upregulation are not fully elucidated. The authors have generated

a Tet-Off transgenic mouse overexpressing P2Y1 selectively in astrocytes (AstroP2Y1OE) and provide evidence indicating that this upregulation enhances neuronal excitability through a signaling pathway involving IGFBP2. Interestingly, they also show that IGFBP2 is upregulated in astrocytes with enhanced expression of P2Y1 in mouse models of stroke and epilepsy.

The article suffers from the fact that the model used is a mouse constitutively overexpressing P2Y1 and therefore the adult phenotype described in this study could result from abnormal developmental trajectories and the bringing into play of compensatory mechanisms. Indeed, these mice have “smaller hippocampi and enlarged ventricles compared with control mice” (end of discussion). The authors do, however, identify a signaling pathway of potential interest in a number of pathologies, but do not fully exploit this finding.

Major points:

1/ Constitutive P2Y1 overexpression. Keeping mice on doxycycline throughout development to avoid transgene expression could have deleterious effects (Inta et al., 2017 *Pediatric Research*), but it would be important to show that stopping overexpression in adults rapidly reverses the phenotype described in the present study.

2/ Validation of IGFBP2 as a potential target for stroke and epilepsy. The authors developed a viral strategy in vivo to knockdown IGFBP2 and they show that they partially inhibit the increased neuronal excitability in AstroP2Y1OE mice (Fig 5). They then show that IGFBP2 is upregulated in P2Y1-expressing astrocytes in models of stroke and epilepsy (Fig 7). However, upregulation of IGFBP2 by reactive astrocytes in a specific pathology may have different consequences from its overexpression in apparently non-reactive astrocytes of AstroP2Y1OE mice. It would therefore be extremely important to test the consequences of knocking down IGFBP2 in models of seizures or stroke.

3/ Calcium responses in astrocytes. Many studies have shown that stimulation of the Schaffer collaterals induces calcium responses in astrocytes of the CA1 stratum radiatum (See for instance Tang et al., 2015 *J Neurosci* for adult rodents). It is therefore extremely surprising to see almost no response in control astrocytes after trains of up to 300 stimulations at 40 Hz (Fig 1D & F). This is most likely the consequence of the use of a very low stimulation current (0.06 mA) that, when reported on the I/O curve shown in Supp Fig 2, is at the very beginning of the curve (about -0.05 mV for the fiber volley and -0.05 mV/ms for the EPSP slope). With these low-intensity stimulations, it is extremely difficult to find the right match between the recording area and the position of the stimulating electrode. Moreover, imaging experiments were carried out on a single optical plane (with unspecified temporal and spatial resolution) and there is a high risk of missing the Ca²⁺ responses in astrocytes. In the absence of combined optical and electrophysiological recordings, how did the authors standardize the procedure between control and AstroP2Y1OE slices to measure Ca²⁺ in astrocytes at the right location? There is a major risk of bias in this type of experiments with unblinded experimenters.

4/ Calcium responses in neurons. The authors claim (lines 199-200 & 216-217) that enhanced neuronal Ca²⁺ responses were mediated by ionotropic glutamatergic receptors. There is no doubt that ionotropic glutamate receptors trigger the responses but their enhancement may well be independent of these receptors but rather due to the increased excitability of neurons shown in

Supp Fig 2. The authors have not ruled out the possibility that neuronal Ca²⁺ signals are partly due to the activation of voltage-gated calcium signals activated by synaptic responses and/or by backpropagating action potentials. Electrophysiological experiments of Supp Fig 2F-H actually show that excitatory synaptic transmission at Schaffer collaterals-CA1 synapses does not differ between control and AstroP2Y1OE mice, thus ruling out the possibility that enhanced neuronal Ca²⁺ responses result from enhanced basal excitatory synaptic transmission. Besides voltage-gated channels, another possibility could be that stimulation trains used for calcium imaging eventually activate extra-synaptic glutamate receptors, most likely NMDA receptors. Interestingly, activation of NMDA receptors can induce neuronal ATP release (Eyo et al., 2014; Dissing-Olesen et al., 2014). Testing the effect of D-AP5 alone (or NR2B-selective antagonists) should help solving this question. The prediction is that blocking only NMDA receptors will do as much (on both neuronal and astrocytic Ca²⁺ responses) as blocking both AMPA and NMDA receptors. It is worth noting that the authors would better use NBQX than CNQX to block non-NMDA receptors, given that CNQX also blocks the glycine site of NMDA receptors.

5/ Glutamate signaling. There is an apparent contradiction between the data of Supp. Fig 2 showing no change in I/O curve at Schaffer Col/CA1 synapses and those of Fig.3 showing with iGluSnFR the enhancement of extracellular glutamate during trains of stimulations at the same synapses. Based on the different kinetics of iGluSnFR and GCaMP6 responses in astrocytes, the authors argue that elevated glutamate during trains is derived from neurons rather than from astrocytes. If we accept this argument, then where does the excess glutamate come from during stimulation trains in P2Y1OE mice? Does it arise from presynaptic potentiation of glutamate release at Schaffer collaterals? If so, it cannot be mediated by mGluRs since blockade of these receptors increases the neuronal Ca²⁺ response (Fig 2B), which is in itself surprising and requires explanation. Or does it come from a higher release of recurrent collaterals of CA1 pyramidal cells that have an enhanced excitability? Or from an abnormal modulation of glutamate transporters during trains (see Armbruster et al., 2016 J Neurosci)? This question needs to be answered because, as things stand, this part of the manuscript is confusing.

6/ IGFBP2 signaling. Fig 5 shows that blocking IGFBP2 with an antibody or knocking it down with a gene editing strategy decreases neuronal Ca²⁺ responses during stimulation trains. This decrease is observed at the level of the soma but not in the dendrites. Does this mean that we are dealing with two different pathways here? Does IGFBP2 knockdown also affect enhanced neuronal excitability (Supp Fig 2) and enhanced glutamate signaling (Fig 3)? Isn't it surprising that blocking IGFBP2 reduced neuronal Ca²⁺ response without affecting astrocytic Ca²⁺ responses (Fig 5C) knowing that blocking ionotropic glutamate receptors inhibited neuronal Ca²⁺ responses and reduced astrocytic Ca²⁺ response (Fig. 2A)?

Other points:

Is there any reason why all experiments were performed on males?

The authors do not specify what control mice are. Did they use wildtype littermates of AstroP2Y1OE?

First paragraph of the results: Isn't it surprising to record interictal spikes in control mice?

The authors should show, on the picture of Fig 1A for instance, the type of ROI they used. It is not clear if ROIs were on single dendrites of neurons or included several dendrites and, for astrocytes if they were on the soma or the processes or both.

What is the temporal resolution of image acquisition?

Did the authors make a cut between CA1 and CA3 for the analysis of synaptic transmission in picrotoxin?

Legend of Supp Fig 3: Not clear. Specify that these are responses to EFS

When describing Fig 2A in the text the authors state "Unlike neuronal Ca²⁺ signals, astrocytic Ca²⁺ signals were largely unaltered by the presence of CNQX/D-AP5" (lines 186-189). The Fig 2A shows rather a reduction of dF/F for astrocytes with a P<0.0001! The reduction is smaller than in neurons but still quite significant. The data on Fig 2A thus suggests that Ca²⁺ responses in astrocytes are due to both pre and postsynaptic release of ATP and this should be clearly stated.

Same remark regarding Fig 2B. The authors claim that MPEP/LY341495 did not affect either neuronal or astrocytic Ca²⁺ signals. The figure shows an increase for neurons with a p=0.00292.

Page 14, Line 201. Fig 2F instead of Fig 2E.

On the same page, the description of Supp Fig 3 is not very precise. The authors do not mention that probenecid also reduced neuronal Ca²⁺ responses and that DCPIP increased neuronal Ca²⁺ responses. It is not clear why they conclude that this analysis suggests that ATP may be released through pannexin channels in neurons. Why not in astrocytes? These drugs are far from being specific and this analysis does not bring much. Of note, the article of Dissing-Olesen et al 2014 showed that neuronal ATP release induced by the activation of NMDARs is not mediated by pannexin.

Throughout the text, Schaffer collaterals instead of Schaffer collateral

Reviewer #3 (Remarks to the Author):

Shigetomi et al. here created a novel mouse line that conditionally overexpresses P2Y_{1R} specifically in astrocytes. They report that freely behaving AstroP2Y₁OE show higher hippocampal spiking rates and lower seizure threshold compared to control mice. Using dual-color two-photon imaging in vitro, they find that astroglial calcium responses are higher in transgenic mice, a finding attributable to ATP release through neuronal pannexin channels. Astrocytic P2Y_{1R} overexpression also increased extracellular glutamate levels, possibly by presynaptic rather than astroglial release. RNAseq revealed an upregulation of Igfbp2, and acute Igfbp2 inhibition neuronal calcium, and recombinant Igfbp2 enhanced it. Moreover, CRISPR/Cas-mediated Igfbp2 knockdown reduced evoked neuronal calcium signals, and this was attributable to IGF-1 receptor actions.

Overall, this is an interesting and relevant study that provides a novel mechanism on how P2Y1R upregulation leads to increased neuronal activity in disease models. The experimental design was thoughtful and most data are convincing, but there are a few points that should be addressed:

1. The Results section immediately starts with the electrophysiological experiments, however it would be good to provide some background on how P2Y1R knockout was achieved. In particular, data should be provided how efficacious this approach is (RNA levels for P2y1r are shown, but protein levels would also be good if possible) and how specific (i.e. is there P2Y1R upregulation outside of astrocytes in transgenic mice either on the protein or mRNA level, e.g. using RNAscope?).

2. The figures lack legibility (e.g. graphs are too small) and contain missing descriptions of groups/axes/scale bars and some typos. Some figures also lack information on groups sizes and statistical methods (e.g. Fig. 7).

3. Were all control mice age-matched littermates?

4. There is no attempt to demonstrate actual ATP release, i.e. the first step in the putative mechanism, although this has become feasible using GRAB-ATP. Alternatively, manipulations to reduce the levels of released ATP, e.g. using apyrase, would be interesting.

5. Is the release of Igfbp2 specific to P2Y1R or generally related to increased calcium levels in astrocytes? Methods to tests this would be by increasing calcium by uncaging or chemogenic astrocyte stimulation (if those methods are available to the authors) and to then measure the effects of IGF-1R blockade.

6. The relatively bold claim that astrocytic glutamate release is not involved in the mechanism is not well back up by the presented data. This is a critical point as P2Y1R has been implicated in glutamate release (e.g. previous data and a recent publication by the Volterra lab). Fig. 3E in particular lacks a statistical analysis. Moreover, the claim is largely based on MRS2365 puff experiments, which are indirect at best. Either the data should be backed up by experiments using blockers of glutamate release (or transgenic mice), or the claim should be significantly toned down.

7. With now many astrocyte datasets published and available, e.g. for Alzheimer's disease and other chronic brain diseases, both for rodent models as well as human tissue, it would be interesting to interrogate these datasets for the upregulation of Igfbp2.

Reviewer #4 (Remarks to the Author):

Reviewer #5 (Remarks to the Author):

RE: NCOMMS-23-38456. “Reactive astrocytes enhance neuronal excitability via IGFBP2: pathological effects of P2Y1 receptor upregulation” by Eiji Shigetomi, Hideaki Suzuki, Yukiho J. Hirayama, Fumikazu Sano, Kohei Yoshihara, Keisuke Koga, Toru Tateoka, Hideyuki Yoshioka, Youichi Shinozaki, Hiroyuki Kinouchi, Kenji F. Tanaka, Haruhiko Bito, Makoto Tsuda & Schuichi Koizumi

Reviewer #1:

The reviewer wrote *“The authors have tested the functional consequences of astrocytic P2Y1 receptor (P2Y1R) upregulation, which has been observed in several pathophysiological conditions. They used an impressive set of imaging, electrophysiological and molecular approaches to dissect the effects of P2Y1R overexpression. They found that P2Y1R overexpression increased seizure susceptibility and neuronal excitability in vivo and in vitro and also increased astrocytic Ca²⁺ signal activity in vitro. The authors further analysed the mechanisms underlying these observations in acute hippocampal slices and also performed glutamate imaging. They further characterised how P2Y1R overexpression affects RNA expression using RNA-seq and identified IGFBP2 as a potential link between P2Y1R overexpression and the observed functional changes. Interestingly, reducing IGFBP2 levels by neutralizing antibodies and reducing astrocytic IGFBP2 expression decreased neuronal activity whereas IGFBP2 incubation increased it. The authors also showed that the increased expression of P2Y1R and IGFBP2 is reproduced in animal models of epilepsy and stroke. These are important and novel findings. In contrast to many previous studies, which have described the many simultaneous changes of astrocyte physiology in disease and their potential implications, this study isolates the effect of one particular relevant change (expression of P2Y1R) and demonstrates how it can reproduce important aspects of disease pathophysiology. The use of P2Y1R overexpression model is elegant, although it does have some significant disadvantages probably due to the persistent P2Y1R upregulation as discussed by the authors.”*

We appreciate the reviewer’s encouraging comments and careful review. Below, we address the reviewer’s specific concerns on a point-by-point basis.

Comment#1-1:

1) The authors dissect the activation of neurons and astrocytes in hippocampal slices after stimulation of Schaffer collateral synapses. They conclude that stimulus-induced astrocytic Ca²⁺ transients require action potential firing and are almost exclusively mediated by P2Y1R receptors. This seems at odds with reports from the same brain region (Tang et al. 2015, Panatier et al. 2011,

among many others). What is the explanation for this discrepancy?

Ans:

Thank you for this insightful comment. We cannot exclude the possible contribution of mGluRs to Ca^{2+} signals. One possibility for the differences may be related to differences in stimulus conditions. To avoid stimulus-induced tissue movement, we imaged at $\sim 250\text{-}300\ \mu\text{m}$ from the stimulus electrode, which is likely quite far away compared with other studies. In addition, we used a relatively low stimulus current (0.06 mA), although it is difficult to compare stimulus intensity between studies because different stimulus electrodes were used. Under our conditions, we did not see large Ca^{2+} signals in controls, whereas robust Ca^{2+} signals were observed in AstroP2Y1OE. Furthermore, ATP imaging data (as suggested by Reviewer 3) showed that the evoked ATP increase—which can activate P2Y1R—was equivalent (**Supplementary Figure 4**). We therefore think that AstroP2Y1OE astrocytes express more P2Y1R molecules, which more effectively respond to the release of ATP to cause larger Ca^{2+} signals. Thus, we believe that these differences in the imaging/stimulus settings are likely to have caused the discrepancies between our and previous results. To clarify this issue, we have added the detailed imaging/stimulus settings to the revised Methods, and have added an explanation of why astrocytic Ca^{2+} signals in the present study are likely caused by P2Y1R activation to the revised Results, as follows.

The lack of clear astrocytic Ca^{2+} signals in the control mice is somewhat inconsistent with previous reports of astrocytic Ca^{2+} signals in the same region of the hippocampus^{41,42}. This discrepancy is likely caused by differences in imaging/stimulus settings; presumably, we used a weaker stimulus intensity than past reports, meaning that astrocytic Ca^{2+} signals in the control mice were very small.

Comment#1-2:

It is stated that astrocytic Ca^{2+} signals are largely unaltered by CNQX/APV (line #186-188) although the reduction in Fig. 2A is highly significant. Based on further experiments the authors conclude that neuronal pannexin channels may release ATP. It is not clear how this may work. What is the trigger of pannexin-mediated ATP release? If it were due to postsynaptic depolarisation/activity it would be highly sensitive to CNQX/APV.

Ans:

We apologize for the confusion; we oversimplified our interpretation of the data. The CNQX/D-AP5 data indicate that P2Y1R activation depends on postsynaptic depolarization. As suggested by Reviewer 2, we performed new experiments with D-AP5 only; we did not find any reduction

of astrocytic Ca^{2+} signals, suggesting that NMDA receptors may not play a major role in ATP release. However, we cannot rule out the possibility that astrocytes release ATP in response to neuronal activity. Our pharmacological experiments suggest that ATP may be released from pannexin or connexin hemichannels. However, the blockade of astrocytic Ca^{2+} signals by probenecid was not 100%, and other pathways for ATP release, such as calcium homeostasis modulator (CALHM) channels, may therefore contribute to ATP release. The way in which activity-dependent ATP release occurs is an important issue; however, it was not the focus of the present study. We have a separate project underway to elucidate the mechanism underlying ATP release, and we hope to follow up on this issue in the near future. To clarify these issues, we have rephrased the text as follows.

Unlike neuronal Ca^{2+} signals, astrocytic Ca^{2+} signals remained but were significantly reduced in the presence of CNQX/D-AP5, suggesting that both presynaptic and postsynaptic activities are required for astrocytic Ca^{2+} signals (**Figure 2A**). The blockade of NMDA receptors with D-AP5 reduced neuronal but not astrocytic Ca^{2+} signals (**Supplementary Figure 3C**), suggesting that NMDA receptor activation may not play a major role in astrocytic Ca^{2+} signals in our experimental conditions.

Although the pharmacological reagents that were used were not entirely specific, the overall pharmacological data suggest that some ATP may be released through pannexin/connexin hemichannels expressed in neurons and/or astrocytes.

Comment#2:

2) Experiments were done using iGluSnFR expressed by astrocytes. (Please state in the manuscript how and which iGluSnFR was expressed.) An increase in iGluSnFR signals was observed in P2Y1R-overexpressing animals. This could reflect an increase of glutamate release, a reduction in glutamate uptake or a shifted balance between glutamate transporter and iGluSnFR expression (see for instance Armbruster et al. 2020). Can the authors distinguish these possibilities, also considering their findings from Suppl. Fig. 2F-h? The absence of mRNA changes of proteins relevant for glutamate homeostasis does not exclude changes of protein function/location/expression and, thereby, functional changes.

Ans:

To address your concerns, we performed slice patch-clamp recordings from CA1 pyramidal neurons and found that the paired-pulse ratio of evoked EPSCs recorded from AstroP2Y1OE slices was lower than that recorded from control slices (**Supplementary Figure 2I–K**). This

suggests that the glutamate release probability is increased in AstroP2Y1OE. An increase in iGluSnFR signals in AstroP2Y1OE mice is therefore caused by increased glutamate release from presynaptic terminals and the subsequent accumulation of extracellular glutamate. Repetitive stimulation (40 Hz, 300 times, 0.06 mA) led to glutamate accumulation, thus causing sustained inward currents. There was a trend toward larger sustained inward currents being recorded from AstroP2Y1OE neurons (-264 ± 33 pA, $n = 12$) than from control neurons (-187 ± 21 pA, $n = 9$), although this trend was not significant ($p = 0.08$). Nonetheless, we cannot rule out the possibility that glutamate uptake may be altered. However, an immunohistochemical analysis of GLT-1 and GLAST (the major glutamate transporters that are expressed in astrocytes) revealed similar expression levels (**Supplementary Figure 5A, B**), consistent with the RNA-seq data.

Because basal excitatory synaptic transmission was not altered but the glutamate release probability was enhanced in AstroP2Y1OE (**Supplementary Figure 2**), the enhanced glutamate iGluSnFR signal is probably due to an increase in glutamate release from presynaptic terminals.

Immunohistochemical analysis of GLT-1 and GLAST revealed similar expression levels in AstroP2Y1OE mice compared to control mice (**Supplementary Figure 5A, B**).

Comment#3:

3) *Why does IGFBP2 affect somatic but not dendritic Ca^{2+} signals of neurons (Fig. 5 and Suppl. Fig. 5) and 6)? If IGFBP2 does not significantly affect dendritic Ca^{2+} signals then it cannot explain the changes observed elsewhere (e.g. Fig. 1C-D), where presumably dendritic Ca^{2+} signals were included in the analysis and quantification since stratum radiatum was named as the region of interest for analysis.*

Ans:

Thank you for your valuable comment. Dendritic Ca^{2+} signals exhibit a biphasic response. The first phase is a fast increase/decay of the Ca^{2+} signal, and the second phase is a sustained Ca^{2+} increase. When we analyzed the first phase of dendritic Ca^{2+} signals, we observed larger Ca^{2+} signals by recombinant IGFBP2. We have now included this analysis in **Supplementary Figure 6B**. Thus, IGFBP2 also acts on dendritic sites, although the effect is small. It remains unknown why the somatic Ca^{2+} signals in neurons were more susceptible to recombinant IGFBP2; however, there might be differences in IGF-1R expression levels between somatic and dendritic sites.

Comment#4:

4) *The results in Fig. 6 are difficult to interpret in the context of P2Y1R overexpression because*

there seems to be no comparison to control mice. If the effect of increasing IGFBP2 in P2Y1R overexpressing animals is due to an interaction with the IGF1 receptor, then the effect of Linsitinib should be different between control and P2Y1R overexpressing animals.

Ans:

Thank you for your important comment. We have included new data (**Figure 6B**) showing the effects of linsitinib on neuronal Ca^{2+} signals in control mice. We did not find any effect of linsitinib on evoked neuronal Ca^{2+} signals from control mice. The data suggest that the IGF-1R-mediated enhancement of evoked neuronal Ca^{2+} signals was observed specifically in AstroP2Y1OE. Although IGFBP2 was expressed in astrocytes in control mice, any contribution of IGFBP2 to evoked neuronal Ca^{2+} signals was not detectable in our experimental setup.

Comment#5:

5) Why did the authors analyse the striatum for the MCAO experiment and not the hippocampus as done elsewhere? What happened in the cortex or hippocampus of the mice in these experiments?

Ans:

We chose the striatum for the analysis because it is one of the major perfusion areas from the MCA. In accordance with your suggestion, however, we also analyzed the cortex and hippocampus. We have added the related data to **Supplementary Figure 8**.

Comment#6:

6) The authors further speculate that IGFBP2 enhances glutamatergic synaptic transmission (line #358-360) but Suppl. Fig. 2F-H and the lack of a clear effect on dendritic Ca^{2+} signals argue against that.

Ans:

Thank you for your valuable comments. Consistent with the data shown in **Supplementary Figure 2F-H**, we did not find any differences in the amplitudes of evoked EPSCs in **Supplementary Figure 2I, J**. Furthermore, neuronal Ca^{2+} signals were larger with repetitive stimuli in **Figure 1**. We therefore think that IGFBP2 increases the release probability of glutamate based on the following findings. First, neuronal Ca^{2+} signals in AstroP2Y1OE but not controls were reduced by an IGF-1 receptor antagonist, suggesting the contribution of IGF-1 receptors to neuronal excitation specifically in AstroP2Y1OE. Based on the literature, IGF-1 receptors are expressed in presynaptic sites of the hippocampus and regulate glutamate release (Gazit et al.,

Neuron, 2016). Second, **Supplementary Figure 2K** shows that the glutamate release probability of CA3–CA1 synapses was higher in AstroP2Y1OE than in controls. Repetitive stimulation may therefore be required for the appearance of an effect of IGFBP2. However, we cannot rule out its effect on postsynaptic sites. We have therefore rephrased our description regarding the site of IGFBP2 action, as follows.

Basal excitatory synaptic transmission was not altered when we measured fEPSP slopes (**Supplementary Figure 2F-H**) and evoked EPSC amplitude (**Supplementary Figure 2J**). The paired-pulse ratio was lower in AstroP2Y1OE (**Supplementary Figure 2K**), suggesting the increase in glutamate release probability by AstroP2Y1OE. Overall, neuronal Ca²⁺ signals in AstroP2Y1OE were larger in response to repetitive inputs of excitatory synapses (**Figure 1D-F**).

Our data indicate that glutamate release is facilitated at CA3–CA1 synapses in response to repetitive stimulation in AstroP2Y1OE mice, thus suggesting a presynaptic effect (**Figures 1D, E and 3, Supplementary Figure 2I–K**). However, the effect of IGFBP2 on neuronal excitation seemed to be larger at the soma (where relatively few excitatory synapses exist) than in dendrites (**Figure 5D, F, Supplementary Figures 6 and 7D**). Exogenous IGFBP2 increases firing and enhances excitatory synaptic transmission at the same synapses via a postsynaptic effect during development⁵⁷. Astrocyte-derived IGFBP2 therefore probably acts via IGF-1R on both presynaptic and postsynaptic sites in neurons—to regulate glutamatergic synaptic transmission and increase firing, respectively—thus leading to hyperexcitability in AstroP2Y1OE mice (**Figure 8**).

Comment#7:

7) The authors mention ,basal glutamate levels‘ in the discussion (line #347-349) but baseline or resting glutamate levels/concentrations have not been reported in the study.

Ans:

Thank you for your comment; we agree. Our iGluSnFR experiments were able to detect differences in extracellular glutamate changes but not in basal glutamate levels. Our data show that P2Y1R activation in AstroP2Y1ROE does not lead to obvious glutamate responses. We apologize for the confusion, and have rephrased the text as follows.

However, our extracellular glutamate imaging findings suggest that P2Y1R activation does not cause obvious glutamate elevations,...

Reviewer #2:

The reviewer wrote “*The study by Shigetomi et al. investigates the consequences of constitutive over-expression of P2Y1 receptors in astrocytes on neuronal and astrocytic functions in the mouse hippocampus. P2Y1 receptors are up-regulated in reactive astrocytes in different neurological disease (stroke, epilepsy, AD...) but the consequences of this upregulation are not fully elucidated. The authors have generated a Tet-Off transgenic mouse overexpressing P2Y1 selectively in astrocytes (AstroP2Y1OE) and provide evidence indicating that this upregulation enhances neuronal excitability through a signaling pathway involving IGFBP2. Interestingly, they also show that IGFBP2 is upregulated in astrocytes with enhanced expression of P2Y1 in mouse models of stroke and epilepsy.*

The article suffers from the fact that the model used is a mouse constitutively overexpressing P2Y1 and therefore the adult phenotype described in this study could result from abnormal developmental trajectories and the bringing into play of compensatory mechanisms. Indeed, these mice have “smaller hippocampi and enlarged ventricles compared with control mice” (end of discussion). The authors do, however, identify a signaling pathway of potential interest in a number of pathologies, but do not fully exploit this finding.”

We appreciate the reviewer’s careful review and valuable comments. Below, we address the reviewer’s specific concerns on a point-by-point basis.

Comment#1:

1/ Constitutive P2Y1 overexpression. Keeping mice on doxycycline throughout development to avoid transgene expression could have deleterious effects (Inta et al., 2017 Pediatric Research), but it would be important to show that stopping overexpression in adults rapidly reverses the phenotype described in the present study.

Ans:

Thank you for your comment. Because we used the Tet-OFF system, we did not use doxycycline to overexpress P2Y1R. To avoid any confusion, we have added this detail to the Methods section.

Comment#2:

2/ Validation of IGFBP2 as a potential target for stroke and epilepsy. The authors developed a viral strategy in vivo to knockdown IGFBP2 and they show that they partially inhibit the increased neuronal excitability in AstroP2Y1OE mice (Fig 5). They then show that IGFBP2 is upregulated in P2Y1-expressing astrocytes in models of stroke and epilepsy (Fig 7). However, upregulation of

IGFBP2 by reactive astrocytes in a specific pathology may have different consequences from its overexpression in apparently non-reactive astrocytes of AstroP2Y1OE mice. It would therefore be extremely important to test the consequences of knocking down IGFBP2 in models of seizures or stroke.

Ans:

Thank you for your comment. We agree that IGFBP2 upregulation in reactive astrocytes in epilepsy or stroke may have different consequences from that in AstroP2Y1OE. We have therefore started experiments relating to IGFBP2 in a stroke model. In this new study, we aim to investigate the spatiotemporal pattern of IGFBP2 and its related molecules (IGF-1, IGF1-R, IGF-2, etc.) to determine the area in the brain that should be targeted for IGFBP2 knockdown. Although we are finding interesting results using this stroke model, it is a new project based on the findings in the present study, and we think that it is out of the present study's scope. We will therefore follow up this issue in our next paper.

Comment#3:

3/ Calcium responses in astrocytes. Many studies have shown that stimulation of the Schaffer collaterals induces calcium responses in astrocytes of the CA1 stratum radiatum (See for instance Tang et al., 2015 J Neurosci for adult rodents). It is therefore extremely surprising to see almost no response in control astrocytes after trains of up to 300 stimulations at 40 Hz (Fig 1D & F). This is most likely the consequence of the use of a very low stimulation current (0.06 mA) that, when reported on the I/O curve shown in Supp Fig 2, is at the very beginning of the curve (about -0.05 mV for the fiber volley and -0.05 mV/ms for the EPSP slope). With these low-intensity stimulations, it is extremely difficult to find the right match between the recording area and the position of the stimulating electrode. Moreover, imaging experiments were carried out on a single optical plane (with unspecified temporal and spatial resolution) and there is a high risk of missing the Ca²⁺ responses in astrocytes. In the absence of combined optical and electrophysiological recordings, how did the authors standardize the procedure between control and AstroP2Y1OE slices to measure Ca²⁺ in astrocytes at the right location? There is a major risk of bias in this type of experiments with unblinded experimenters.

Ans:

Thank you for your valuable comment. This issue was also raised by Reviewer 1. We agree that we used a relatively low stimulation current, which was chosen to avoid movement artifacts. In our experiments, we carefully decided where to image rather than looking for a field of view that showed responses to the stimuli. We used specific regions of the hippocampus (approximately

-1.6 to -2.3 mm from the bregma) and imaged ~250-300 μm from the stimulus electrode, which may be relatively far compared with other studies. We therefore think that these differences in imaging/stimulus settings may have caused the discrepancies between our study and past studies. Although our stimulus current was relatively low, we were able to detect neuronal Ca^{2+} signals and EPSCs. We apologize that we did not include image acquisition information in the original manuscript. We have added the detailed imaging/stimulus settings to the revised Methods, and have added an explanation of why astrocytic Ca^{2+} signals in the present study are likely caused by P2Y1R activation to the revised Results, as follows.

The lack of clear astrocytic Ca^{2+} signals in the control mice is somewhat inconsistent with previous reports of astrocytic Ca^{2+} signals in the same region of the hippocampus^{41,42}. This discrepancy is likely caused by differences in imaging/stimulus settings; presumably, we used a weaker stimulus intensity than past reports, meaning that astrocytic Ca^{2+} signals in the control mice were very small.

Comment#4:

4/ Calcium responses in neurons. The authors claim (lines 199-200 & 216-217) that enhanced neuronal Ca^{2+} responses were mediated by ionotropic glutamatergic receptors. There is no doubt that ionotropic glutamate receptors trigger the responses but their enhancement may well be independent of these receptors but rather due to the increased excitability of neurons shown in Supp Fig 2. The authors have not ruled out the possibility that neuronal Ca^{2+} signals are partly due to the activation of voltage-gated calcium signals activated by synaptic responses and/or by backpropagating action potentials. Electrophysiological experiments of Supp Fig 2F-H actually show that excitatory synaptic transmission at Schaffer collaterals-CA1 synapses does not differ between control and AstroP2Y1OE mice, thus ruling out the possibility that enhanced neuronal Ca^{2+} responses result from enhanced basal excitatory synaptic transmission. Besides voltage-gated channels, another possibility could be that stimulation trains used for calcium imaging eventually activate extra-synaptic glutamate receptors, most likely NMDA receptors. Interestingly, activation of NMDA receptors can induce neuronal ATP release (Eyo et al., 2014; Dissing-Olesen et al., 2014). Testing the effect of D-AP5 alone (or NR2B-selective antagonists) should help solving this question. The prediction is that blocking only NMDA receptors will do as much (on both neuronal and astrocytic Ca^{2+} responses) as blocking both AMPA and NMDA receptors. It is worth noting that the authors would better use NBQX than CNQX to block non-NMDA receptors, given that CNQX also blocks the glycine site of NMDA receptors.

Ans:

Thank you for your valuable comment and proposal. As suggested, we have performed experiments using D-AP5 alone; we found that D-AP5 inhibited neuronal but not astrocytic Ca^{2+} signals. The observation of neuronal Ca^{2+} signals in the presence of NMDA receptors (shown in **Supplementary Figure 3C**) indicates that Ca^{2+} signals in dendrites are partly caused by the activation of voltage-gated Ca^{2+} channels. We have added this interpretation to the revised Results. Furthermore, Supplementary Figure 2F–H suggests that basal excitatory synaptic transmission may not be altered in AstroP2Y1OE. However, Supplementary Figure 2I–K shows that the release probability of glutamate was enhanced in AstroP2Y1OE. We therefore think that repetitive stimulation accumulates more glutamate in AstroP2Y1OE, which then activates AMPA/NMDA receptors, including extrasynaptic NMDA receptors. The absence of an effect of D-AP5 on astrocytic Ca^{2+} signals suggests that, under our experimental conditions, NMDA receptors may not play a major role in ATP release.

Comment#5:

5/ Glutamate signaling. There is an apparent contradiction between the data of Supp. Fig 2 showing no change in I/O curve at Schaffer Col/CA1 synapses and those of Fig.3 showing with iGluSnFR the enhancement of extracellular glutamate during trains of stimulations at the same synapses. Based on the different kinetics of iGluSnFR and GCaMP6 responses in astrocytes, the authors argue that elevated glutamate during trains is derived from neurons rather than from astrocytes. If we accept this argument, then where does the excess glutamate come from during stimulation trains in P2Y1OE mice? Does it arise from presynaptic potentiation of glutamate release at Schaffer collaterals? If so, it cannot be mediated by mGluRs since blockade of these receptors increases the neuronal Ca^{2+} response (Fig 2B), which is in itself surprising and requires explanation. Or does it come from a higher release of recurrent collaterals of CA1 pyramidal cells that have an enhanced excitability? Or from an abnormal modulation of glutamate transporters during trains (see Armbruster et al., 2016 J Neurosci)? This question needs to be answered because, as things stand, this part of the manuscript is confusing.

Ans:

Thank you; this concern was also raised by Reviewer 1. To address this point, we performed slice patch-clamp recordings from CA1 pyramidal neurons and found that the paired-pulse ratio of evoked EPSCs recorded from AstroP2Y1OE slices was lower than that of control slices (**Supplementary Figure 2I–K**). This suggests that the glutamate release probability is increased in AstroP2Y1OE. An increase in iGluSnFR signals in AstroP2Y1OE mice is therefore caused by increased glutamate release from presynaptic terminals and the subsequent accumulation of extracellular glutamate. Repetitive stimulation (40 Hz, 300 times, 0.06 mA) led to glutamate

accumulation, thus causing sustained inward currents. There was a trend toward larger sustained inward currents being recorded from AstroP2Y1OE neurons (-264 ± 33 pA, $n = 12$) than from control neurons (-187 ± 21 pA, $n = 9$), although this trend was not significant ($p = 0.08$). Nonetheless, we cannot rule out the possibility that glutamate uptake may be altered. However, an immunohistochemical analysis of GLT-1 and GLAST (the major glutamate transporters that are expressed in astrocytes) revealed similar expression levels (**Supplementary Figure 5A, B**), consistent with the RNA-seq data.

Basal excitatory synaptic transmission was not altered when we measured fEPSP slopes (**Supplementary Figure 2F-H**) and evoked EPSC amplitude (**Supplementary Figure 2J**). The paired-pulse ratio was lower in AstroP2Y1OE (**Supplementary Figure 2K**), suggesting the increase in glutamate release probability by AstroP2Y1OE. Overall, neuronal Ca^{2+} signals in AstroP2Y1OE were larger in response to repetitive inputs of excitatory synapses (**Figure 1D-F**).

Because basal excitatory synaptic transmission was not altered but the glutamate release probability was enhanced in AstroP2Y1OE (**Supplementary Figure 2**), the enhanced glutamate iGluSnFR signal is probably due to an increase in glutamate release from presynaptic terminals.

Immunohistochemical analysis of GLT-1 and GLAST revealed similar expression levels in AstroP2Y1OE mice compared to control mice (**Supplementary Figure 5A, B**).

Comment#6:

6/ IGFBP2 signaling. Fig 5 shows that blocking IGFBP2 with an antibody or knocking it down with a gene-editing strategy decreases neuronal Ca^{2+} responses during stimulation trains. This decrease is observed at the level of the soma but not in the dendrites. Does this mean that we are dealing with two different pathways here? Does IGFBP2 knockdown also affect enhanced neuronal excitability (Supp Fig 2) and enhanced glutamate signaling (Fig 3)? Isn't it surprising that blocking IGFBP2 reduced neuronal Ca^{2+} response without affecting astrocytic Ca^{2+} responses (Fig 5C) knowing that blocking ionotropic glutamate receptors inhibited neuronal Ca^{2+} responses and reduced astrocytic Ca^{2+} response (Fig. 2A)?

Ans:

Thank you for your valuable comments. The sites of action of IGFBP2 may be both the soma and dendrites. Khan et al. (*Adv Sci*, 2019) reported that exogenous IGFBP2 increases CA1 pyramidal neuron firing and enhances excitatory synaptic transmission. At dendritic sites, we found that the first phase of neuronal Ca^{2+} signals was significantly reduced by IGFBP2 knockdown in

astrocytes. We have added this analysis as **Supplementary Figure 7D**. In addition, exogenous IGFBP2 enhanced the first phase of dendritic Ca²⁺ signals, thus indicating that IGFBP2 enhances dendritic Ca²⁺ signals, at least in part. It remains unknown why the soma is more susceptible to IGFBP2 knockdown or treatment; however, there might be differences in the sites/expression levels of IGF-1R between somatic and dendritic sites. Notably, it was somewhat surprising that we did not see any effects of IGFBP2 antibodies on astrocytic Ca²⁺ signals. This may be because of differences in the reduction of postsynaptic neuronal Ca²⁺ responses between CNQX/D-AP5 and IGFBP2 antibodies. The inhibitory effect on neuronal Ca²⁺ signals of CNQX/DAP-5 (60%) was much larger than that of the IGFBP2 antibody (29%).

Our data indicate that glutamate release is facilitated at CA3–CA1 synapses in response to repetitive stimulation in AstroP2Y1OE mice, thus suggesting a presynaptic effect (**Figures 1D, E and 3, Supplementary Figure 2I–K**). However, the effect of IGFBP2 on neuronal excitation seemed to be larger at the soma (where relatively few excitatory synapses exist) than in dendrites (**Figure 5D, F, Supplementary Figures 6 and 7D**). Exogenous IGFBP2 increases firing and enhances excitatory synaptic transmission at the same synapses via a postsynaptic effect during development⁵⁷. Astrocyte-derived IGFBP2 therefore probably acts via IGF-1R on both presynaptic and postsynaptic sites in neurons—to regulate glutamatergic synaptic transmission and increase firing, respectively—thus leading to hyperexcitability in AstroP2Y1OE mice (**Figure 8**).

Comment#7:

Is there any reason why all experiments were performed on males?

Ans:

There was no specific reason for using male mice. We wanted to avoid the effects of sex differences on data variation, and we chose male mice because male mice have been used in most past literature.

Comment#8:

The authors do not specify what control mice are. Did they use wildtype littermates of AstroP2Y1OE?

Ans:

We used wild-type (P2ry1tetO knock-in) littermate mice as the control.

Comment#9:

First paragraph of the results: Isn't it surprising to record interictal spikes in control mice?

Ans:

It was indeed surprising to us that we recorded these spikes in control mice. Similarly, however, Purtell et al. (*Epilepsy Behav.* 2018) reported that epileptiform spikes were detected in 100% of C57BL/6J mice, although the biological meaning of these spikes remains unclear.

Comment#10:

The authors should show, on the picture of Fig 1A for instance, the type of ROI they used. It is not clear if ROIs were on single dendrites of neurons or included several dendrites and, for astrocytes if they were on the soma or the processes or both.

Ans:

Thank you; we have added the ROIs to the figure. We used rectangular ROIs therefore they were on several dendrites for neurons and soma and/or processes for astrocytes.

Comment#11:

What is the temporal resolution of image acquisition?

Ans:

We imaged at 3 frames/s; we have added this information to the revised manuscript.

Comment#12:

Did the authors make a cut between CA1 and CA3 for the analysis of synaptic transmission in picrotoxin?

Ans:

No, we did not make a cut between CA1 and CA3 for the analysis.

Comment#13:

Legend of Supp Fig 3: Not clear. Specify that these are responses to EFS

Ans:

Thank you for this comment. We have added more descriptions to the revised figure legend.

Comment#14:

When describing Fig 2A in the text the authors state “Unlike neuronal Ca²⁺ signals, astrocytic Ca²⁺ signals were largely unaltered by the presence of CNQX/D-AP5” (lines 186-189). The Fig 2A shows rather a reduction of dF/F for astrocytes with a P<0.0001! The reduction is smaller than in neurons but still quite significant. The data on Fig 2A thus suggests that Ca²⁺ responses in astrocytes are due to both pre and postsynaptic release of ATP and this should be clearly stated. Same remark regarding Fig 2B. The authors claim that MPEP/LY341495 did not affect either neuronal or astrocytic Ca²⁺ signals. The figure shows an increase for neurons with a p=0.00292.

Ans:

This issue was also noted by Reviewer 1. We apologize for the confusion; we oversimplified our interpretation of the data. The CNQX/D-AP5 data indicate that P2Y1R activation depends on postsynaptic depolarization. However, we cannot rule out the possibility that astrocytes release ATP in response to neuronal activity. We have therefore described that astrocytic Ca²⁺ signals are caused by presynaptic/postsynaptic and astrocytic release of ATP. We also agree with your point regarding the MPEP/LY341495 experiments. In the revised manuscript, we have described the results more precisely.

Comment#15:

Page 14, Line 201. Fig 2F instead of Fig 2E.

Ans:

Thank you for noticing this error; we have corrected it.

Comment#16:

On the same page, the description of Supp Fig 3 is not very precise. The authors do not mention that probenecid also reduced neuronal Ca²⁺ responses and that DCPIP increased neuronal Ca²⁺ responses. It is not clear why they conclude that this analysis suggests that ATP may be released through pannexin channels in neurons. Why not in astrocytes? These drugs are far from being specific and this analysis does not bring much. Of note, the article of Dissing-Olesen et al 2014 showed that neuronal ATP release induced by the activation of NMDARs is not mediated by pannexin.

Ans:

This issue was also raised by Reviewer 1. We agree with your comments, and have rephrased the descriptions of the pharmacological experiments in **Supplementary Figure 3**. As you have noted,

the drugs that we used may not have been specific to the targets that we were focused on, although we chose the concentrations carefully. Of the drugs, probenecid had the strongest effect on EFS-evoked astrocytic Ca^{2+} signals compared with EFS-evoked neuronal Ca^{2+} responses. For this reason, we think that pannexin channels may contribute to the ATP release pathway. However, we cannot rule out the possibility that probenecid may inhibit as-yet unidentified ATP release pathways, as reported by Dissing-Olssen et al. (*J Neurosci*, 2014); it may also inhibit neuronal excitation, thus reducing glutamate release. We also cannot rule out the possibility that astrocytes release ATP through pannexin/connexin channels. We have therefore revised our description of the data to be more precise in the revised manuscript. Interestingly, although DCPIB did not affect astrocytic Ca^{2+} signals, it slightly but significantly enhanced EFS-evoked Ca^{2+} signals in neurons. We are unsure of why this occurred. Recent publications show that DCPIB-sensitive VRAC channel activation may release glutamate from astrocytes (Beppu et al. *J Physiol*, 2021; Yang et al. *Neuron*, 2019). However, DCPIB blocks glutamate release via VRAC, thus inhibiting excitatory synaptic transmission, which is the opposite to what we found. In the revised manuscript, we have therefore rephrased our description of the ATP release mechanism, and have discussed other possible release sites/pathways.

Ans:

Throughout the text, Schaffer collaterals instead of Schaffer collateral

Thank you; we have corrected this term throughout the revised manuscript.

Reviewer #3:

The reviewer wrote “*Shigetomi et al. here created a novel mouse line that conditionally overexpresses P2Y1R specifically in astrocytes. They report that freely behaving AstroP2Y1OE show higher hippocampal spiking rates and lower seizure threshold compared to control mice. Using dual-color two-photon imaging in vitro, they find that astroglial calcium responses are higher in transgenic mice, a finding attributable to ATP release through neuronal pannexin channels. Astrocytic P2Y1R overexpression also increased extracellular glutamate levels, possibly by presynaptic rather than astroglial release. RNAseq revealed an upregulation of Igfbp2, and acute Igfbp2 inhibition neuronal calcium, and recombinant Igfbp2 enhanced it. Moreover, CRISPR/Cas-mediated Igfbp2 knockdown reduced evoked neuronal calcium signals, and this was attributable to IGF-1 receptor actions.*

Overall, this is an interesting and relevant study that provides a novel mechanism on how P2Y1R upregulation leads to increased neuronal activity in disease models. The experimental design was

thoughtful and most data are convincing, but there are a few points that should be addressed.”

We appreciate the reviewer’s encouraging comments and careful review. Below, we address the reviewer’s specific concerns on a point-by-point basis.

Comment#1:

1. The Results section immediately starts with the electrophysiological experiments, however it would be good to provide some background on how P2Y1R knockout was achieved. In particular, data should be provided how efficacious this approach is (RNA levels for *P2y1r* are shown, but protein levels would also be good if possible) and how specific (i.e. is there P2Y1R upregulation outside of astrocytes in transgenic mice either on the protein or mRNA level, e.g. using RNAscope?).

Ans:

Thank you for your valuable comment. We have previously reported RNA levels using *in situ* hybridization (Shigetomi et al. *J Neurosci*, 2018) as shown below; *P2ry1* transcripts were highly upregulated in GFAP-positive astrocytes. We have added a description of our previous finding to the revised manuscript. We have also performed immunohistochemistry for P2Y1R; P2Y1R immunofluorescence was indeed higher in AstroP2Y1OE mice (**Supplementary Figure 1A**).

Figure Redacted due to 3rd Party Rights Concerns

To achieve astrocytic P2Y1R overexpression, we used a Tet-OFF system. We crossed an Mlc1-tTA line (an astrocytic tTA line^{35,36}) with a P2Y1tetO line. We have previously reported that *P2ry1* transcripts are highly upregulated in GFAP-positive astrocytes of AstroP2Y1OE mice by in situ hybridization³⁷. To confirm that protein levels are also upregulated, we performed immunohistochemistry. Indeed, P2Y1R immunofluorescence was significantly higher in GFAP-positive astrocytes in the hippocampal CA1 region of AstroP2Y1OE mice (**Supplementary Figure 1A**).

Comment#2:

2. *The figures lack legibility (e.g. graphs are too small) and contain missing descriptions of groups/axes/scale bars and some typos. Some figures also lack information on groups sizes and statistical methods (e.g. Fig. 7).*

Ans:

Thank you for your comments. As suggested, we have made changes to add missing information and correct all typos in the figures and their legends.

Comment#3:

3. *Were all control mice age-matched littermates?*

Ans:

Yes, we used age-matched littermates for all experiments except that shown in Figure 6 (linsitinib).

Comment#4:

4. *There is no attempt to demonstrate actual ATP release, i.e. the first step in the putative mechanism, although this has become feasible using GRAB-ATP. Alternatively, manipulations to reduce the levels of released ATP, e.g. using apyrase, would be interesting.*

Ans:

Thank you for your valuable comment. We performed ATP imaging using the GRAB_{ATP1.0} sensor and found that the overall ATP dynamics in response to electrical stimuli of the Schaffer collaterals were equivalent between AstroP2Y1OE and controls, although there was a slight increase in ATP elevation in response to stimuli. These findings indicate that differences in astrocytic Ca²⁺ responses between AstroP2Y1OE and controls are therefore mainly the result of differences in P2Y1 receptor expression in astrocytes, rather than ATP levels. We have added the new data

(Supplementary Figure 4) and related descriptions to the revised manuscript.

Comment#5:

5. Is the release of Igfbp2 specific to P2Y1R or generally related to increased calcium levels in astrocytes? Methods to tests this would be by increasing calcium by uncaging or chemogenic astrocyte stimulation (if those methods are available to the authors) and to then measure the effects of IGF-1R blockade.

Ans:

We agree that the way in which IGFBP2 is released is an important issue. We believe that IGFBP2 is upregulated by P2Y1R signaling rather than IGF1R release, although our data do not rule out the possibility of Ca²⁺-dependent IGFBP2 secretion. It would be interesting to investigate whether calcium uncaging or chemogenetic astrocyte stimulation increases IGFBP2 secretion. Using these methods, we would be able measure IGFBP2 secretion. We also think that it is important to visualize whether IGFBP2 is incorporated into vesicles or secretory granules, to understand IGFBP2 secretion mechanisms. However, although we agree that this is an important issue, it is outside the scope of the present study. We will explore this issue in future studies.

Comment#6:

6. The relatively bold claim that astrocytic glutamate release is not involved in the mechanism is not well back up by the presented data. This is a critical point as P2Y1R has been implicated in glutamate release (e.g. previous data and a recent publication by the Volterra lab). Fig. 3E in particular lacks a statistical analysis. Moreover, the claim is largely based on MRS2365 puff experiments, which are indirect at best. Either the data should be backed up by experiments using blockers of glutamate release (or transgenic mice), or the claim should be significantly toned down.

Ans:

Thank you for your suggestion. Figure 3E shows that P2Y1R activation leads to robust Ca²⁺ signals in astrocytes but no elevation of glutamate levels. Furthermore, EFS-evoked glutamate responses terminated after EFS, whereas astrocytic Ca²⁺ signals peaked at/after EFS. These data suggest that P2Y1R activation may not elevate glutamate levels; however, we cannot rule out the possibility that glutamate transients may increase via P2Y1R activation in astrocytes. We also cannot exclude the possibility that a subpopulation of astrocytes releases glutamate via P2Y1R activation. We have therefore rephrased this idea as follows.

This treatment increased Ca^{2+} but did not elevate glutamate near astrocytes (**Figure 3E**), indicating that the instantaneous activation of P2Y1Rs in astrocytes does not elevate glutamate levels under our experimental conditions. However, we cannot rule out the possibility of Ca^{2+} -dependent glutamate release from astrocytes⁵³. Overall, these findings suggest that P2Y1R-overexpressing astrocytes regulate glutamatergic signals partly by facilitating glutamate release from presynaptic terminals.

Comment#7:

7. *With now many astrocyte datasets published and available, e.g. for Alzheimer's disease and other chronic brain diseases, both for rodent models as well as human tissue, it would be interesting to interrogate these datasets for the upregulation of Igfbp2.*

Ans:

Thank you for the suggestion. Using iDEP.96 (<http://bioinformatics.sdstate.edu/idep96/>), a web application, and/or another web application from OlvTools (<https://olvtools.com/>), we have analyzed publicly available databases of Alzheimer's disease models (GSE183050, GSE129797, GSE138695), a Parkinson disease model (GSE191131), and stroke model (GSE103783). However, we did not find changes in *Igfbp2/IGFBP2* or *P2ry1/P2RY1* transcripts. The absence of upregulation of the genes is unknown but may be due to differences in age, strain, and severity of the disease models. Also, we found a relatively large variability in data among biological replicates in some of the datasets, which could be one of reasons for the absence of the upregulation. We would not include this analysis in the revised manuscript. We will follow up expression of *Igfbp2* and *P2ry1* in other disease models in the near future.

Reviewer #4:

We appreciate the reviewer's careful review and valuable comments.

Reviewer #5:

We appreciate the reviewer's careful review and valuable comments.

REVIEWERS' COMMENTS

Reviewer #1 (Remarks to the Author):

The authors have comprehensively and convincingly addressed all my previous points. I have no further comments.

Reviewer #2 (Remarks to the Author):

The authors have performed additional experiments and addressed most of the reviewers' concerns. Nevertheless, with regards to constitutive overexpression of P2Y1, the question was related to the fact that with the Tet-OFF system without doxycycline there is a constitutive overexpression of P2Y1 in astrocytes probably during much of the development, which may induce lasting changes in neuronal and astrocytic networks. Interestingly, IGFBP2 regulates the development of synapses and neurons (Kahn et al., 2019, cited by the authors). It would therefore have been important to test whether doxycycline given to adult P2Y1OE mice reverses the phenotype described by the authors, in particular the calcium responses in both astrocytes and neurons.

Furthermore, the hypothesis that IGFBP2 increases the probability of glutamate release at presynaptic terminals in P2Y1OE mice is not strongly supported by the reported results. The increase in glutamate release measured with iGluSnFR in response to 100 stimuli was small, as was the decrease in PPR observed. If the probability of release were increased, the I/O curve at these synapses should change, which is not the case (Supp. Fig. 2 F-H). The authors should temper this conclusion and discuss other alternatives (see Glasgow et al., Front Synaptic Neurosci. 2019 for instance).

Please correct the legend of Figure 5C. The first sentence makes no sense.

Reviewer #3 (Remarks to the Author):

My previous comment from the first version, which asked if Igfbp2 is upregulated in publicly available datasets, was only addressed in the response letter, but not in the revised manuscript itself. The

finding that the authors did not find changes in Igfbp2 or P2ry1 expression in these datasets should briefly be reported and discussed in the paper.

Other than that, all of my previous points have been sufficiently addressed.

Reviewer #4 (Remarks to the Author):

I co-reviewed this manuscript with one of the reviewers who provided the listed reports. This is part of the Nature Communications initiative to facilitate training in peer review and to provide appropriate recognition for Early Career Researchers who co-review manuscripts

Reviewer #5 (Remarks to the Author):

I co-reviewed this manuscript with one of the reviewers who provided the listed reports. This is part of the Nature Communications initiative to facilitate training in peer review and to provide appropriate recognition for Early Career Researchers who co-review manuscript

RE: NCOMMS-23-38456. “Reactive astrocytes enhance neuronal excitability via IGFBP2: pathological effects of P2Y1 receptor upregulation” by Eiji Shigetomi, Hideaki Suzuki, Yukiho J. Hirayama, Fumikazu Sano, Yuki Nagai, Kohei Yoshihara, Keisuke Koga, Toru Tateoka, Hideyuki Yoshioka, Youichi Shinozaki, Hiroyuki Kinouchi, Kenji F. Tanaka, Haruhiko Bito, Makoto Tsuda & Shuichi Koizumi

Reviewer #2:

Comment#1:

The authors have performed additional experiments and addressed most of the reviewers' concerns. Nevertheless, with regards to constitutive overexpression of P2Y1, the question was related to the fact that with the Tet-OFF system without doxycycline there is a constitutive overexpression of P2Y1 in astrocytes probably during much of the development, which may induce lasting changes in neuronal and astrocytic networks. Interestingly, IGFBP2 regulates the development of synapses and neurons (Kahn et al., 2019, cited by the authors). It would therefore have been important to test whether doxycycline given to adult P2Y1OE mice reverses the phenotype described by the authors, in particular the calcium responses in both astrocytes and neurons.

Ans:

Thank you for your comment. As the reviewer suggested, we cannot rule out the effect of P2Y1 receptor overexpression in astrocytes in development. We have discussed this issue in the original version of the manuscript and have revised it as follows. Although we cannot rule out the possibility of astrocytic P2Y1 receptor overexpression altering neuronal and astrocytic networks, the fact that astrocytic Igfbp2 knockdown or IGFBP2 antibody treatment reduces neuronal excitation in brain slices from adult mice suggests that IGFBP2 in the adult mediate the enhancement of neuronal excitation in AstroP2Y1OE.

In our model, P2Y1Rs were overexpressed throughout all stages of development. We therefore cannot exclude the possibility that P2Y1R overexpression in astrocytes alters neuronal circuits and astrocytic networks in development. However, both the genetic and pharmacological inhibition of IGFBP2 signals reduced neuronal Ca²⁺ signals in brain slices from adult mice, indicating that astrocyte-derived IGFBP2 regulates neuronal Ca²⁺ signals in adults.

Comment#2:

Furthermore, the hypothesis that IGFBP2 increases the probability of glutamate release at

presynaptic terminals in P2Y1OE mice is not strongly supported by the reported results. The increase in glutamate release measured with iGluSnFR in response to 100 stimuli was small, as was the decrease in PPR observed. If the probability of release were increased, the I/O curve at these synapses should change, which is not the case (Supp. Fig. 2 F-H). The authors should temper this conclusion and discuss other alternatives (see Glasgow et al., Front Synaptic Neurosci. 2019 for instance).

Ans:

Thank you for your comment. We agree that iGluSnFR data and I/O data may not strongly support our hypothesis that IGFBP2 increases the probability of glutamate release. As the reviewer suggested, we have revised the Results and Discussion for other alternatives as follows.

Overall, these findings suggest that P2Y1R-overexpressing astrocytes regulate glutamatergic signals partly by facilitating glutamate release from presynaptic terminals, although other mechanisms to enhance glutamatergic transmission may also play a role (See Discussion below).

Although PPR data in **Supplementary Figure 2I-K** suggest an increase in the probability of glutamate release, we did not find a change in the input-output curve (**Supplementary Figure 2F-H**), which may not strongly support the idea of the increase in the release probability. Another possibility is that IGFBP2 inhibits IGF signaling by sequestering IGF from its receptors⁵⁸, which may lead to increased extracellular glutamate levels via reduced IGF-1 signaling in astrocytes⁷⁹. Given that IGFBP2 alters the structure of synapses⁶¹, astrocytic coverage of synapses may change to modulate glutamatergic transmission because the proximity of astrocyte leaflets affects glutamate diffusion and spillover⁸¹. Those changes could also contribute to the enhancement of glutamatergic transmission by IGFBP2.

Comment#3:

Please correct the legend of Figure 5C. The first sentence makes no sense.

Ans:

Thank you for the comment. We have corrected it.

Reviewer #3:

Comment#1:

“My previous comment from the first version, which asked if Igfbp2 is upregulated in publicly available datasets, was only addressed in the response letter, but not in the revised manuscript itself. The finding that the authors did not find changes in Igfbp2 or P2ry1 expression in these datasets should briefly be reported and discussed in the paper.”

Ans:

As the reviewer suggested, we include the description of the analysis of publicly available datasets in the revised manuscript.